# Solving Inverse Problems via Diffusion Optimal Control

**Henry Li** [*]
Yale University
henry.li@yale.edu

**Marcus Pereira**
Bosch Center for Artificial Intelligence
marcus.pereira@us.bosch.com

## Abstract

Existing approaches to diffusion-based inverse problem solvers frame the signal recovery task as a probabilistic sampling episode, where the solution is drawn from the desired posterior distribution. This framework suffers from several critical drawbacks, including the intractability of the conditional likelihood function, strict dependence on the score network approximation, and poor $\mathbf{x}_0$ prediction quality. We demonstrate that these limitations can be sidestepped by reframing the generative process as a discrete optimal control episode. We derive a diffusion-based optimal controller inspired by the iterative Linear Quadratic Regulator (iLQR) algorithm. This framework is fully general and able to handle any differentiable forward measurement operator, including super-resolution, inpainting, Gaussian deblurring, nonlinear deblurring, and even highly nonlinear neural classifiers. Furthermore, we show that the idealized posterior sampling equation can be recovered as a special case of our algorithm. We then evaluate our method against a selection of neural inverse problem solvers, and establish a new baseline in image reconstruction with inverse problems[1].

## 1 Introduction

Diffusion models Song and Ermon [2019], Ho et al. [2020] have been shown to be remarkably adept at conditional generation tasks Dhariwal and Nichol [2021], Ho and Salimans [2022], in part due to their iterative sampling algorithm, which allows the dynamics of an uncontrolled prior score function $\nabla_{\mathbf{x}} \log p_t(\mathbf{x})$ to be directed towards an arbitrary posterior distribution by introducing an additive guidance term $\mathbf{u}$. When this guidance term is the conditional score $\nabla_{\mathbf{x}} \log p_t(\mathbf{y}|\mathbf{x})$, the resulting sample is provably drawn from the desired conditional distribution $p(\mathbf{x}|\mathbf{y})$ Song et al. [2020].

A central obstacle to this framework is the general difficulty of obtaining the conditional score function $\nabla_{\mathbf{x}} \log p_t(\mathbf{y}|\mathbf{x}_t)$ due to its dependence on the *noisy* diffusion variate $\mathbf{x}_t$ rather than just the final sample $\mathbf{x}_0$ Chung et al. [2023a]. In large-scale conditional generation tasks such as class- or text-conditional sampling the computational overhead of training a time-dependent conditional score function from scratch is deemed acceptable, and is indeed the approach taken by Rombach et al. [2022], Saharia et al. [2022], and many others. However, this solution is not acceptable in inverse problems where the goal is to design a generalized solver that will work in a zero-shot capacity for an arbitrary forward model.

This bottleneck has spawned a flurry of recent research dedicated to approximating the conditional score $\nabla_{\mathbf{x}} \log p_t(\mathbf{y}|\mathbf{x}_t)$ as a simple function of the *noiseless* likelihood $\log p(\mathbf{y}|\mathbf{x}_0)$ Choi et al. [2021], Chung et al. [2022], Rout et al. [2024], Chung et al. [2023a], Kawar et al. [2022], Chung et al. [2023b]. However, as we will demonstrate in this work, these approximations impose a significant cost to the performance of the resulting algorithm.

---

[*]Work partially completed during an internship at Bosch AI.

[1]Code is available at https://github.com/lihenryhfl/diffusion_optimal_control.

38th Conference on Neural Information Processing Systems (NeurIPS 2024).

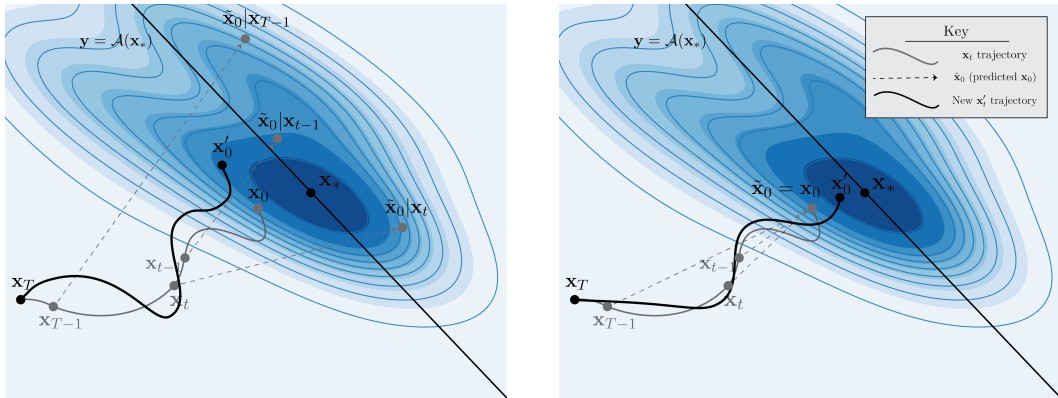

Figure 1: **Conceptual illustration comparing a probabilistic posterior sampler to our proposed optimal control-based sampler.** In a probabilistic sampler, the model relies on an approximation $\tilde{\mathbf{x}}_0 \approx \mathbf{x}_0$ to guide each step **(left)**. We are able to compute $\mathbf{x}_0$ **exactly** on each step, resulting in much higher quality gradients $\nabla \log p(\mathbf{y}|\tilde{\mathbf{x}}_0)$ and an improved trajectory update **(right)**.

To address these issues, we propose a novel framework built from optimal control theory where such approximations are no longer necessary. By framing the reverse diffusion process as an optimal control episode, we are able to detach the inverse problem solver from the strict requirements of the conditional sampling equation given by Song et al. [2020], while still leveraging the exceptionally powerful prior of the unconditional diffusion process. Moreover, we find that the desired score function directly arises as the Jacobian of the value function.

We summarize our contributions as follows:

- We present diffusion optimal control, a framework for solving inverse problems via the lens of optimal control theory, using pretrained unconditional off-the-shelf diffusion models.

- We show that this perspective overcomes many core obstacles present in existing diffusion-based inverse problem solvers. In particular, the idealized posterior sampling score Song et al. [2021] — approximated by existing methods — can be recovered exactly as a specific case of our method.

- We showcase the advantages of our model empirically with quantitative experiments and qualitative examples, and demonstrate state-of-the-art performance on the FFHQ $256 \times 256$ dataset.

## 2 Background

**Notation** We use lowercase letters for denoting scalars $a \in \mathbb{R}$, lowercase bold letters for vectors $\mathbf{a} \in \mathbb{R}^n$ and uppercase bold letters for matrices $\mathbf{A} \in \mathbb{R}^{m \times n}$. Subscripts indicate Jacobians and Hessians of scalar functions, e.g. $l_\mathbf{x} \in \mathbb{R}^n$ and $l_{\mathbf{xx}} \in \mathbb{R}^{n \times n}$ for $l(\mathbf{x}) : \mathbb{R}^n \to \mathbb{R}$, respectively. We overload notation for time-dependent variables, where subscripts imply dependence rather than derivatives w.r.t. time, e.g., $\mathbf{x}_t = \mathbf{x}(t)$. Furthermore, $V(\mathbf{x}_t)$ and $Q(\mathbf{x}_t, \mathbf{u}_t)$ are scalar functions despite being uppercase, in line with existing optimal control literature Betts [1998].

### 2.1 Diffusion Models

The diffusion modeling literature uses the following reverse-time Itö SDE to generate samples Song et al. [2021],

$$\mathrm{d}\mathbf{x}_t = \left[\mathbf{f}(\mathbf{x}_t) - g(t)^2 \nabla_{\mathbf{x}_t} \log p_t(\mathbf{x}_t)\right]\mathrm{d}t + g(t)\mathrm{d}\mathbf{w}_t, \tag{1}$$

where $\mathbf{x}_t \in \mathbb{R}^n$ is the state vector, $\mathbf{f} : \mathbb{R}^n \to \mathbb{R}^n$ and $g : \mathbb{R} \to \mathbb{R}$ are drift and diffusion terms that can take different functional forms (e.g., Variance-Preserving SDEs (VPSDEs) and Variance-Exploding SDEs (VESDEs) in Song et al. [2021]), $\nabla_{\mathbf{x}_t} \log p_t(\mathbf{x}_t)$ is the score-function and $\mathbf{w}_t \in \mathbb{R}^n$ is a vector of mutually independent Brownian motions. The above SDE has an associated ODE called the

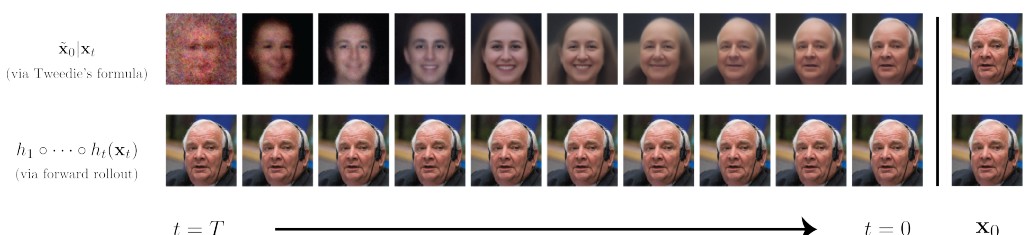

Figure 2: **Predicted $\mathbf{x}_0$ used in a probabilistic framework (above) compared to ours (below) for a general diffusion trajectory.** The full forward rollout in our proposed framework allows for the predicted $\mathbf{x}_0$ (and therefore $\nabla_{\mathbf{x}_t} \log p(\mathbf{y}|\mathbf{x}_0)$) to be efficiently computed for all $t = 0, \ldots, T$.

probability-flow (PF) ODE given by

$$\mathrm{d}\mathbf{x}_t = \mathrm{d}\mathbf{x}_t + \left[ \mathbf{f}(\mathbf{x}_t) - \frac{1}{2}g(t)^2 \nabla_{\mathbf{x}_t} \log p_t(\mathbf{x}_t) \right] \mathrm{d}t, \tag{2}$$

with the same marginals $p_t(\mathbf{x}_t)$ as the SDE, which allow for likelihood computation [Song et al., 2021, Li et al., 2024]. All practical implementations of diffusion samplers require a time-discretization of the PF-ODE. One such discretization is the well-known Euler-discretization which gives,

$$\mathbf{x}_{t-1} = \mathbf{x}_t - \left[ \mathbf{f}(\mathbf{x}_t) - \frac{1}{2}g(t)^2 \nabla_{\mathbf{x}} \log p_t(\mathbf{x}_t) \right] \Delta t \tag{3}$$

where, $\Delta t$ is the length of the discretization interval and we have reversed the time evolution by changing the sign of the drift. We are not restricted to only using the Euler-discretization and any high-order discretization techniques can also be employed. More concisely, we have,

$$\mathbf{x}_{t-1} = \mathbf{h}(\mathbf{x}_t), \quad \text{where } \mathbf{h} : \mathbb{R}^n \to \mathbb{R}^n \tag{4}$$

which describes the general non-linear dynamics of the corresponding discrete-time diffusion sampler.

## 2.2 Posterior Sampling for Inverse Problems

Inverse problems are a general class of problems where an unknown signal is reconstructed from observations obtained by a forward measurement process Ongie et al. [2020]. The forward process is usually lossy, resulting in an ill-posed signal recovery task where a *unique* solution does not exist. The forward model can generally be written as

$$y = \mathcal{A}(\mathbf{x}_0) + \eta, \tag{5}$$

where $\mathcal{A} : \mathbb{R}^n \to \mathbb{R}^d$ is the forward operator, $y \in \mathbb{R}^d$ the measured signal, $\mathbf{x}_0 \in \mathbb{R}^n$ the unknown signal to be recovered, and $\eta \sim \mathcal{N}(0, \sigma \mathbf{I}_d)$ the noise (with variance $\sigma^2$) in the measurement process.

Given the forward model Eq. (5) and a measurement $\mathbf{y}$, sampling from the posterior distribution $p_\theta(\mathbf{x}|\mathbf{y})$ can then be performed by solving the corresponding *conditional* Itô SDE

$$\mathrm{d}\mathbf{x} = [\mathbf{f}(\mathbf{x}) - g(t)^2 \nabla_{\mathbf{x}} \log p_t(\mathbf{x}|\mathbf{y})]\mathrm{d}t + g(t)\mathrm{d}\mathbf{w}, \tag{6}$$

where, invoking Bayes rule,

$$\nabla_{\mathbf{x}} \log p_t(\mathbf{x}|\mathbf{y}) = \nabla_{\mathbf{x}} \log p_t(\mathbf{x}) + \nabla_{\mathbf{x}} \log p_t(\mathbf{y}|\mathbf{x}). \tag{7}$$

As with the unconditional dynamics, Eq. (6) has a corresponding ODE

$$\mathrm{d}\mathbf{x} = [\mathbf{f}(\mathbf{x}) - \frac{1}{2}g(t)^2 \nabla_{\mathbf{x}} \log p_t(\mathbf{x}|\mathbf{y})]\mathrm{d}t, \tag{8}$$

which has an approximate solution obtained by the Euler discretization

$$\mathbf{x}_{t-1} = \mathbf{x}_t + [f(\mathbf{x}_t) - \frac{1}{2}g(t)^2 \nabla_{\mathbf{x}_t} \log p_t(\mathbf{x}_t|\mathbf{y})]\Delta t. \tag{9}$$

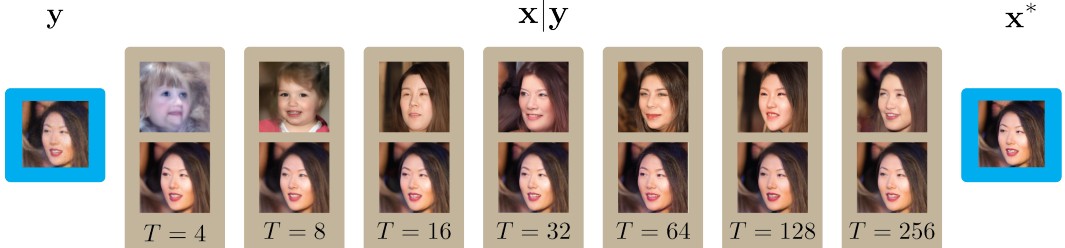

Figure 3: **Inverse problem solution as a function of total diffusion timesteps $T$ for the $4\times$ super-resolution task.** Compared to DPS (**top row**), our method (**bottom row**) produces solutions that are higher quality, in greater agreement with the inverse problem contraint $\mathcal{A}\mathbf{x} = \mathbf{y}$, and more stable across $T$.

## 2.3 Optimal Control

Optimal control is the structured and principled approach to the guidance of dynamical systems over time. Many methods have been developed in the optimal control literature and are popularly referred to as *trajectory optimization* algorithms Betts [1998]. Perhaps the most well-known is the Iterative Linear Quadratic Regulator (iLQR) algorithm which uses a first-order approximation of the dynamics and second-order approximations of the value-function Li and Todorov [2004].

Formally, let us define an arbitrary user-defined global cost function

$$J_T = \sum_{t=T}^{1} \ell_t(\mathbf{x}_t, \mathbf{u}_t) + \ell_0(\mathbf{x}_0), \tag{10}$$

composed of a sum over scalar-valued *running* and *terminal* cost functions $\ell_t$ and $\ell_0$. Optimal control theory dictates that the value function $V(\mathbf{x}_t, t) := \min_{\{\mathbf{u}_n\}_{n=t}^{n=1}} J_t$ satisfies the following recursive relation also known as *Bellman's Principle of Optimality*

$$V(\mathbf{x}_t, t) = \min_{\mathbf{u}_t} \left[ \ell_t(\mathbf{x}_t, \mathbf{u}_t) + V(\mathbf{x}_{t-1}, t-1) \right]. \tag{11}$$

The iLQR algorithm centers around approximating the state-action value function,

$$Q(\mathbf{x}_t, \mathbf{u}_t) := \ell_t(\mathbf{x}_t, \mathbf{u}_t) + V(\mathbf{x}_{t-1}, t-1), \tag{12}$$

from which the value function can be recovered as $V(\mathbf{x}_t, t) = \min_{\mathbf{u}_t} Q(\mathbf{x}_t, \mathbf{u}_t)$.

Then given a state transition function $\mathbf{x}_t = \mathbf{h}(\mathbf{x}_{t+1}, \mathbf{u}_{t+1})$ where we crucially note that we have defined time to flow *backwards* from $t = T, \ldots, 0$, the iLQR algorithm has feedforward and feedback gains

$$\mathbf{k} = -Q_{\mathbf{uu}}^{-1} Q_{\mathbf{u}} \qquad \text{and} \qquad \mathbf{K} = -Q_{\mathbf{uu}}^{-1} Q_{\mathbf{ux}} \tag{13}$$

The update equations can be written as

$$V_{\mathbf{x}} = Q_{\mathbf{x}} - \mathbf{K}^T Q_{\mathbf{uu}} \mathbf{k} \qquad \text{and} \qquad V_{\mathbf{xx}} = Q_{\mathbf{xx}} - \mathbf{K}^T Q_{\mathbf{uu}} \mathbf{K}. \tag{14}$$

Given the feedforward and feedback gains $\{(\mathbf{K}_t, \mathbf{k}_t)\}_{t=0}^{T}$ and $\bar{\mathbf{x}}_0 := \mathbf{x}_0$, we can recursively obtain the locally optimal control at time $t$ as a function of the present states $\mathbf{x}_t$ and controls $\mathbf{u}_t$ as

$$\bar{\mathbf{x}}_t = \mathbf{h}(\bar{\mathbf{x}}_{t+1}, \mathbf{u}_{t+1}^*), \tag{15}$$

$$\mathbf{u}_t^* = \mathbf{u}_t + \lambda \mathbf{k} + \mathbf{K}(\bar{\mathbf{x}}_t - \mathbf{x}_t). \tag{16}$$

For a more detailed treatment of iLQR as well as a derivation of the equations, please see Appendix B.

## 3 Diffusion Optimal Control

We motivate our framework by observing that the reverse diffusion process Eq. (1) is an uncontrolled non-linear dynamical system that evolves from some initial state (at time $t = T$) to some terminal state (at time $t = 0$). By injecting control vectors $\mathbf{u}_t$ into this system we can influence its behavior and hence its terminal state (i.e., the generated data) to sample from a desired $p(\mathbf{x}|\mathbf{y})$. There are two obvious ways to inject control into this process:

---

**Algorithm 1** Diffusion Optimal Control

---
**Input:** $\lambda, T, \mathbf{y}, \mathbf{x}_T$
**Initialize** $\mathbf{u}_t, \mathbf{k}_t, \mathbf{K}_t$ as $\mathbf{0}$ for $t = 1 \ldots T$, $\{\mathbf{x}'_t\}_{t=0}^T$ as uncontrolled dynamics
**for** iter $= 1$ **to** num_iters **do**
    $V_{\mathbf{x}}, V_{\mathbf{xx}} \leftarrow \nabla_{\mathbf{x}_0} \log p(\mathbf{y}|\mathbf{x}_0), \nabla_{\mathbf{x}_0}^2 \log p(\mathbf{y}|\mathbf{x}_0)$               $\triangleright$ **Initialize derivatives of** $V(\mathbf{x}_t, t)$
    **for** $t = 1$ **to** $T$ **do**
        **Compute** $\mathbf{k}_t, \mathbf{K}_t, V_{\mathbf{x}}, V_{\mathbf{xx}}$                       $\triangleright$ **See Eqs.** (13), (14)
    **end for**
    **for** $t = T$ **to** $1$ **do**
        $\mathbf{x}_{t-1} \leftarrow h(\mathbf{x}_t, \lambda \mathbf{k}_t + \mathbf{K}_t(\mathbf{x}_t - \mathbf{x}'_t))$           $\triangleright$ **Update** $\mathbf{x}_{t-1}$ **with new** $\mathbf{u}_t$
        $\mathbf{x}'_t \leftarrow \mathbf{x}_t$
    **end for**
**end for**

---

1. In **input perturbation control**, we apply the $\mathbf{u}_t$ *before* the diffusion step:

$$\mathbf{x}_{t-1} = (\mathbf{x}_t + \mathbf{u}_t) - \left[\mathbf{f}(\mathbf{x}_t + \mathbf{u}_t) - \frac{1}{2}g(t)^2 \nabla_{\mathbf{x}} \log p_t(\mathbf{x}_t + \mathbf{u}_t)\right]\Delta t. \tag{17}$$

2. In **output perturbation control**, $\mathbf{u}_t$ is applied *after* the diffusion step:

$$\mathbf{x}_{t-1} = \mathbf{x}_t - \left[\mathbf{f}(\mathbf{x}_t) - \frac{1}{2}g(t)^2 \nabla_{\mathbf{x}} \log p_t(\mathbf{x}_t)\right]\Delta t + \mathbf{u}_t. \tag{18}$$

Observe that iLQR is formulated for general discrete-time dynamic processes. When applied specifically to the reverse diffusion dynamics of diffusion models, we are able to make several simplifications. First, we assume that we do not have access to any guidance except at time $t = 0$ — i.e., $\ell_t(\mathbf{x}_t, \mathbf{u}_t)$ does not depend on $\mathbf{x}_t$.

In the case of **input perturbation control**, we observe from Eq. (17) that $\mathbf{h}_{\mathbf{x}} = \mathbf{h}_{\mathbf{u}}$, whereas **output perturbation control** implies that $\mathbf{h}_{\mathbf{u}} = \mathbf{I}$, resulting in the left and right equations, respectively:

$$Q_{\mathbf{x}} = \mathbf{h}_{\mathbf{x}}^T V'_{\mathbf{x}} \qquad\qquad\qquad Q_{\mathbf{x}} = \mathbf{h}_{\mathbf{x}}^T V'_{\mathbf{x}} \tag{19}$$

$$Q_{\mathbf{u}} = \ell_{\mathbf{u}} + \mathbf{h}_{\mathbf{x}}^T V'_{\mathbf{x}} \qquad\qquad Q_{\mathbf{u}} = \ell_{\mathbf{u}} + V'_{\mathbf{x}} \tag{20}$$

$$Q_{\mathbf{xx}} = \mathbf{h}_{\mathbf{x}}^T V'_{\mathbf{xx}} \mathbf{h}_{\mathbf{x}} \qquad\qquad Q_{\mathbf{xx}} = \mathbf{h}_{\mathbf{x}}^T V'_{\mathbf{xx}} \mathbf{h}_{\mathbf{x}} \tag{21}$$

$$Q_{\mathbf{ux}} = Q_{\mathbf{xu}} = \mathbf{h}_{\mathbf{x}}^T V'_{\mathbf{xx}} \mathbf{h}_{\mathbf{x}} \qquad\qquad Q_{\mathbf{ux}} = Q_{\mathbf{xu}}^T = V'_{\mathbf{xx}} \mathbf{h}_{\mathbf{x}} \tag{22}$$

$$Q_{\mathbf{uu}} = \ell_{\mathbf{uu}} + \mathbf{h}_{\mathbf{x}}^T V'_{\mathbf{xx}} \mathbf{h}_{\mathbf{x}} \qquad\qquad Q_{\mathbf{uu}} = \ell_{\mathbf{uu}} + V'_{\mathbf{xx}}. \tag{23}$$

The derivatives of $V$ can then be backpropagated using the following equations:

$$V_{\mathbf{x}} = Q_{\mathbf{x}} - \mathbf{K}^T Q_{\mathbf{uu}} \mathbf{k} = Q_{\mathbf{xx}} - \mathbf{K}^T Q_{\mathbf{uu}} \mathbf{K}$$
$$= Q_{\mathbf{x}} + Q_{\mathbf{ux}}^T Q_{\mathbf{uu}}^{-1} Q_{\mathbf{u}} \tag{24}$$
$$V_{\mathbf{xx}} = Q_{\mathbf{xx}} - \mathbf{K}^T Q_{\mathbf{uu}} \mathbf{K}$$
$$= Q_{\mathbf{xx}} - Q_{\mathbf{ux}}^T Q_{\mathbf{uu}}^{-1} Q_{\mathbf{ux}}. \tag{25}$$

In high dimensional systems such as Eq. 3, matrices may be singular. Therefore, a Tikhonov regularized variant of iLQR is often employed, where matrix inverses are regularized by a diagonal matrix $\alpha \mathbf{I}$ Tassa et al. [2014].

## 3.1 High Dimensional Control

Compared to the dynamics in traditional application areas of optimal control, those we consider in Eqs. (17- 18) are much higher dimensional in the state $\mathbf{x}$ and control $\mathbf{u}$ variates. Therefore, iLQR faces several unique computational bottlenecks when applied to such control problems.

In particular, the Jacobian matrices $\mathbf{h}_{\mathbf{x}}, \mathbf{h}_{\mathbf{u}}$ and the second-order derivative matrices $V_{\mathbf{xx}}, Q_{\mathbf{xx}}, Q_{\mathbf{ux}}, Q_{\mathbf{xu}}$, and $Q_{\mathbf{uu}}$ are particularly expensive to compute, store, and perform downstream operations against. For example, in a three-channel $256 \times 256$ image, these matrices naively contain $(256 \times 256 \times 3)^2 \approx 39B$ parameters.

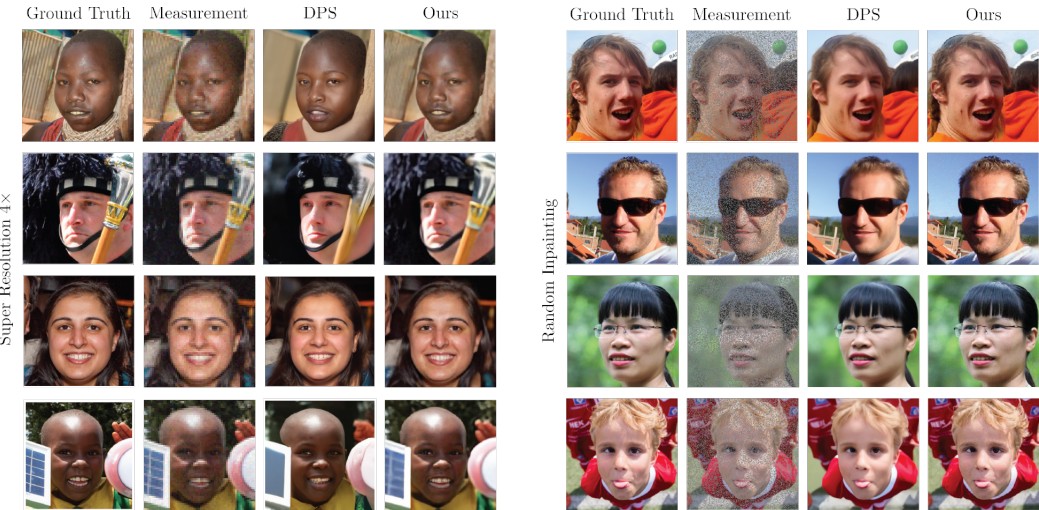

Figure 4: **Examples from inverse problem tasks on FFHQ** $256 \times 256$. From left to right each column contains ground truth, measurement, Diffusion Posterior Sampling (DPS), and ours.

In Appendix D.1 we propose and analyze three modifications to the standard iLQR algorithm: **randomized low rank approximations**, **matrix-free evaluations**, and action updates via an **adaptive optimizer**, that significantly reduce runtime and memory constraints while introducing minimal deterioration to performance on inverse problem solving tasks.

## 4    Improved Posterior Sampling

We demonstrate that our optimal control-based sampler overcomes several practical obstacles that plague existing diffusion-based methods for inverse problem solvers.

**Brittleness to Discretization**    In a probabilistic framework, solutions to inverse problems incur a discretization error from the numerical solution of Eq. (8) that decays poorly with the total diffusion steps $T$ of the diffusion process. While much research has been conducted on the acceleration of unconditional diffusion processes Song et al. [2020], Jolicoeur-Martineau et al. [2021], Karras et al. [2022], Meng et al. [2023], sample quality appears to decay much more aggressively in diffusion-based inverse problem solvers (Figure 3).

We theorize that this is due to two reasons: 1) the posterior sampler Eq. (9) is only correct in the limit of infinitely small time steps, and 2) the quality of the approximated conditional score term $\nabla_\mathbf{x} \log p(\mathbf{y}|\mathbf{x}_t)$ decays quickly with time (Figure 2), and so fewer timesteps lead to fewer chances at low $t$ to correct errors made at high $t$. On the other hand, since optimal control directly casts the **discretized** process as an end-to-end control episode, it produces a feasible solution for any number of discretization steps $T$.

**Intractability of** $\nabla_{\mathbf{x}_t} \log p(\mathbf{y}|\mathbf{x}_t)$    When the forward model $\mathcal{A}$ is known and $\eta$ comes from a simple distribution, the conditional likelihood $p(\mathbf{y}|\mathbf{x}_t)$ can be derived in closed form for $t = 0$. On the other hand, the dependence of $y$ on $\mathbf{x}_t$ for $t > 0$ is generally not known without explicitly computing $\mathbf{x}_0$, which requires sampling from the diffusion process. Ultimately, obtaining the conditional score term $\nabla_{\mathbf{x}_t} \log p(\mathbf{y}|\mathbf{x}_t)$ is a highly nontrivial task Song et al. [2021].

To sidestep this issue, many works Meng and Kabashima [2022], Song et al. [2022], Chung et al. [2023a] factorize this term as the integral

$$p(\mathbf{y}|\mathbf{x}_t) = \int p(\mathbf{y}|\mathbf{x}_0)p(\mathbf{x}_0|\mathbf{x}_t)d\mathbf{x}_0 \tag{26}$$

and then apply a series of approximations to recover a computationally feasible estimate of the conditional score. First, the marginal $p(\mathbf{x}_0|\mathbf{x}_t)$ is replaced by the marginal conditioned on $\mathbf{x}_0$, i.e.

$p(\mathbf{x}_0|\mathbf{x}_t, \mathbf{x}_0) = \mathcal{N}(\mathbf{x}_0, \sigma^2 \mathbf{I})$ Kim and Ye [2021]. Next, the $\mathbf{x}_0$-centered marginal is replaced by the posterior mean $\mathbb{E}[\mathbf{x}_0|\mathbf{x}_t]$ given by Tweedie's formula Efron [2011]. Finally, the true score is replaced by the learned score network.

While these approximations are necessary in a probabilistic framework, we show that they are not required in our method. Intuitively, this is because the linear quadratic regulator backpropagates the control cost $\log p(\mathbf{y}|\mathbf{x})$ through a forward trajectory rollout, which naturally computes the true conditional score at each time $t$. Moreover, our model always estimates $\mathbf{x}_0|\mathbf{x}_t$ exactly (up to the discretization error induced by solving Eq. 3), rather than forming an approximation $\hat{\mathbf{x}}_0 \approx \mathbf{x}_0$ (Figure 2). We formalize this observation with the following statement.

**Theorem 4.1.** *Let Eq. 3 be the discretized sampling equation for the diffusion model with **output perturbation mode** control (Eq. 18). Moreover, let the terminal cost*

$$\ell_0(\mathbf{x}_0) = -\log p(\mathbf{y}|\mathbf{x}_0) \tag{27}$$

*be twice-differentiable and the running costs*

$$\ell_t(\mathbf{x}_t, \mathbf{u}_t) = 0. \tag{28}$$

*Then the iterative linear quadratic regulator with Tikhonov regularizer $\alpha$ produces the control*

$$\mathbf{u}_t = \alpha \nabla_{\mathbf{x}_t} \log p(\mathbf{y}|\mathbf{x}_0). \tag{29}$$

In other words, by framing the inverse problem as an unconditional diffusion process with controls $\mathbf{u}_t$, our proposed method produces controls that coincide precisely with the desired conditional scores $\nabla_{\mathbf{x}_t} \log p(\mathbf{y}|\mathbf{x}_0)$.

Let us further assume that $\log p(\mathbf{y}|\mathbf{x}_t) = \log p(\mathbf{y}|\mathbf{x}_0)$, i.e., $\mathbf{x}_t$ contains no additional information about $y$ than $\mathbf{x}_0$. This assumption results in the posterior mean approximation in Chung et al. [2023a] under stochastic dynamics (Eq. 1), where we additionally obtain *exact* computation of $\mathbf{x}_0$, rather than $\hat{\mathbf{x}}_0 \approx \mathbf{x}_0$ via Tweedie's formula Kim and Ye [2021]. Under the deterministic ODE dynamics (Eq. 2), we recover the **true posterior sampler** under appropriate choice of Tikhonov regularization constant $\alpha$.

**Lemma 4.2.** *Under the deterministic sampler with **output perturbation mode** control, $\alpha = \frac{1}{g(t)^2 \Delta t}$ recovers posterior sampling (Eq. 9).*

We demonstrate a similar result with **input mode perturbation**.

**Theorem 4.3.** *Let Eq. 3 be the discretized sampling equation for the diffusion model with **input perturbation mode** control (Eq. 17). Moreover, let*

$$\ell_0(\mathbf{x}_0) = \log p(\mathbf{y}|\mathbf{x}_0), \tag{30}$$

*and the running costs*

$$\ell_t(\mathbf{x}_t, \mathbf{u}_t) = 0. \tag{31}$$

*Then the iterative linear quadratic regulator with Tikhonov regularizer $\alpha = \frac{1}{g(t)^2 \Delta t}$ produces the dynamical sytem*

$$\widetilde{\mathbf{x}}_t = \widetilde{\mathbf{x}}_t + [f(\widetilde{\mathbf{x}}_t) - \frac{1}{2}g(t)^2(\nabla_{\mathbf{x}} \log p_t(\widetilde{\mathbf{x}}_t)$$
$$+ \nabla_{\mathbf{x}} \log p_t(\mathbf{y}|\mathbf{x}_t))]\Delta t, \tag{32}$$

*where $\widetilde{\mathbf{x}}_t := \mathbf{x}_t + \mathbf{u}_t$.*

Observe that Eq. (32) can be understood as a predictor-corrector sampling method, where the predictor produces an unconditional reverse diffusion update and the corrector produces a conditional correction step on the intermediary variable $\mathbf{x}_t = \tilde{\mathbf{x}}_t - \mathbf{u}_t$.

Ultimately, these results demonstrate that our proposed method is able to recover the idealized sampling procedure under mild assumptions on the diffusion optimal control algorithm.

| | SR $\times 4$ | | Random Inpainting | | Box Inpainting | | Gaussian Deblurring | | Motion Deblurring | |
|---|---|---|---|---|---|---|---|---|---|---|
| | FID $\downarrow$ | LPIPS $\downarrow$ | FID $\downarrow$ | LPIPS $\downarrow$ | FID $\downarrow$ | LPIPS $\downarrow$ | FID $\downarrow$ | LPIPS $\downarrow$ | FID $\downarrow$ | LPIPS $\downarrow$ |
| Ours (NFE = 2500) | **32.47** | **0.171** | **15.93** | **0.053** | **20.22** | **0.122** | **31.80** | **0.189** | **39.40** | **0.217** |
| Ours (NFE = 1000) | 37.53 | 0.189 | 20.75 | 0.108 | 23.88 | 0.164 | 35.24 | 0.191 | 45.99 | 0.233 |
| PSLD (NFE = 1000) | 34.28 | 0.201 | 21.34 | 0.096 | 43.11 | 0.167 | 41.53 | 0.221 | - | - |
| Flash-Diffusion* (NFE = *varies*) | - | - | 53.95 | 0.195 | - | - | 65.35 | 0.280 | 64.57 | 0.267 |
| DDNM (NFE = 1000) | 68.94 | 0.328 | 105.3 | 0.802 | 72.28 | 0.483 | 126.0 | 0.995 | - | - |
| DPS (NFE = 1000) | 39.35 | 0.214 | 33.12 | 0.168 | 21.19 | 0.212 | 44.05 | 0.257 | 39.92 | 0.242 |
| DDRM (NFE = 1000) | 62.15 | 0.294 | 42.93 | 0.204 | 69.71 | 0.587 | 74.92 | 0.332 | - | - |
| MCG (NFE = 1000) | 87.64 | 0.520 | 40.11 | 0.309 | 29.26 | 0.286 | 101.2 | 0.340 | 310.5 | 0.702 |
| PNP-ADMM | 66.52 | 0.353 | 151.9 | 0.406 | 123.6 | 0.692 | 90.42 | 0.441 | 89.08 | 0.405 |
| Score-SDE (NFE = 1000) | 96.72 | 0.563 | 60.06 | 0.331 | 76.54 | 0.612 | 109.0 | 0.403 | 292.2 | 0.657 |
| ADMM-TV | 110.6 | 0.428 | 68.94 | 0.322 | 181.5 | 0.463 | 186.7 | 0.507 | 152.3 | 0.508 |

Table 1: Quantitative evaluation (FID, LPIPS) of model performance on inverse problems on the FFHQ 256x256-1K dataset.

**Dependence on the Approximate Score**    While our theoretical results require that the learned score function $s_\theta(\mathbf{x}_t, t)$ approximates the true data score $\log p_t(\mathbf{x}_t, t)$, we emphasize that the performance of our method does not necessitate this condition. In fact, we find that reconstruction performance is theoretically and empirically robust to the accuracy of the approximated prior score $s_\theta(\mathbf{x}_t, t) \approx \nabla_{\mathbf{x}_t} \log p_t(\mathbf{x}_t)$ or conditional score $\nabla_{\mathbf{x}_t} \log p_t(\mathbf{y}|\mathbf{x}_0) \approx \nabla_{\mathbf{x}_t} \log p_t(\mathbf{y}|\mathbf{x}_t)$ terms. This is because the optimal control-based solution is formulated for the optimization of generalized dynamical systems, and thus agnostic to the diffusion sampling process.

Certainly, improved approximation of the score terms result in a better-informed prior and usually higher sample quality. However, we demonstrate that our sampler produces remarkably reasonable solutions even in the case of randomly initialized diffusion models. Conversely, probabilistic posterior samplers can only sample from $p(\mathbf{y}|\mathbf{x}_0)$ when the terms composing the posterior sampling equation (Eq. (8)) are well approximated (Figure 6). Modeling errors can occur even in foundation models. For example, this scenario may arise in models trained on regions where there are underrepresented examples in the data. When these arise from existing social or ethical biases, they can further perpetuate or amplify biases to the resulting model if left unaddressedBolukbasi et al. [2016], Birhane et al. [2021], Srivastava et al. [2022].

There exist several methods that seek to alleviate the errors incurred by Tweedie's formula (being a mean approximation of the diffusion process), including Song et al. [2024] which imposes a hard data consistency optimization loop at various points in the diffusion process, and Rout et al. [2023] which includes a stochastic averaging loop in each step of the diffusion process. However, these methods still rely on Tweedie's formula for the error reduction scheme, which assumes access to a ground truth score function. Ultimately, the aforementioned problems in the present section are exacerbated in existing samplers, and relatively less consequential in our solver.

# 5   Related Work

The recent success of diffusion models in image generation Song and Ermon [2019], Ho et al. [2020], Song et al. [2021], Rombach et al. [2022] has spawned a surge of research in deep learning-based solvers to inverse problems. Song et al. [2021] demonstrated a strategy for provably sampling from the solution set $p(\mathbf{x}|\mathbf{y})$ of a general inverse problem $\mathbf{y} = \mathcal{A}(\mathbf{x})$ using only an unconditional prior score model $\nabla_{\mathbf{x}} \log p_t(\mathbf{x})$ and a forward probabilistic model $\log p(\mathbf{y}|\mathbf{x}_t)$. However, a crucial problem arises in the intractability of forward probabilistic model, which depends on the noisy $\mathbf{x}_t$ rather than the final $\mathbf{x}_0$. This has resulted in a series of approximation algorithms Choi et al. [2021], Kawar et al. [2022], Chung et al. [2022, 2023a,b], Kawar et al. [2023] for the true conditional diffusion dynamics.

Topics in control theory have been applied to deep learning Liu et al. [2020], Pereira et al. [2020] as well as diffusion modeling Berner et al. [2022]. Optimal control can also be connected to diffusion processes via forward-backward SDEs Chen et al. [2021]. However, these ideas have not been applied to guided conditional diffusion processes solely at inference time, nor for guided conditional sampling. Our proposed optimal control-based algorithm is, to our knowledge, the first such framework for deep inverse problem solvers.

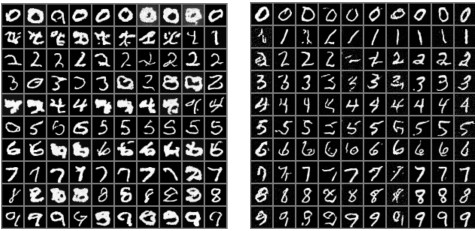

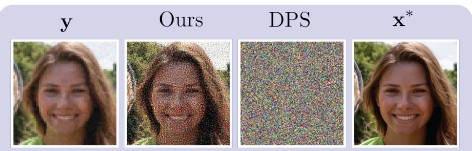

Figure 5: **Examples from the class-conditional inverse problem.** DPS (left) is compared against ours (right). Each row is a different target MNIST class.

Figure 6: **Robustness to approximation quality of the score function.** We consider the $4\times$ super-resolution task with a *randomly initialized* diffusion model. Since the reverse diffusion process is no longer well approximated, DPS cannot produce a feasible solution, while our method still can.

## 6  Experiments

Following previous work Chung et al. [2023a], Meng and Kabashima [2022], Kawar et al. [2022], we consider five inverse problems. 1) In $4\times$ image super-resolution, we use the bicubic downsampling operator. 2) In randomized inpainting, we uniformly omit 92% of all pixels (across all channels). 3) In box inpainting, we mask out a $128 \times 128$ block uniformly sampled from a 16 pixel margin from each side of the image, as in Chung et al. [2022]. 4) In Gaussian deblurring, we use a kernel of size $61 \times 61$ and standard deviation 3.0. In motion deblurring, we generate images according to a library[2] of point spread functions with kernel size $61 \times 61$ and intensity 0.5. Following the experimental design in Chung et al. [2023a], we apply Gaussian noise with standard deviation 0.05 to all measurements of the forward model.

We compare against a generalized diffusion inverse sampler (Score-SDE) proposed in Song et al. [2021], Diffusion Posterior Sampling (DPS) Chung et al. [2023a], Denoising Diffusion Restoration Models Kawar et al. [2022], Manifold Constrained Gradients (MCG) Chung et al. [2022], as well as two recent latent diffusion-based methods Fabian et al. [2023] (Flash-Diffusion[3]) and Rout et al. [2024] (PSLD). For non-diffusion baselines, we compare against Plug-and-Play Alternating Direction Method of Multipliers (PnP-ADMM) with neural proximal maps Chan et al. [2016], Zhang et al. [2017], and a total-variation based alternating direction method of multipliers (TV-ADMM) baseline proposed in Chung et al. [2023a].

We validate our results on the high resolution human face dataset FFHQ $256 \times 256$ Karras et al. [2019]. Several methods are model agnostic (DPS, DDRM, MCG, and thus evaluated with the same pre-trained diffusion models. To fairly compare between all models, all methods use the model weights from Chung et al. [2023a], which are trained on 49K FFHQ images, with 1K images left as a held-out set for evaluation. We compare our algorithm against competing frameworks on these last 1K images. We report our results on FFHQ $256 \times 256$ in Table 1, and demonstrate improvements on all tasks against previous methods. Finally, we demonstrate the performance of our algorithm on the nonlinear inverse problem of class-conditional generation. Namely, let $\mathcal{A}(\mathbf{x}) = \texttt{classifier}(\mathbf{x})$ and $p(\mathbf{y}|\mathbf{x})$ be its associated probability. We compare our method to DPS on the inverse task of generating an MNIST digit given a label $\mathbf{y}$. Compared to images generated by DPS, images from our method exhibit more pronounced class alignment and higher overall sample quality (Figure 5).

## 7  Conclusion

In this paper we presented a novel perspective on tackling inverse problems with diffusion models – framing the discretized reverse diffusion process as a discrete time optimal control episode. We demonstrate that this framework alleviates several core problems in probabilistic solvers: its dependence on the approximation quality of the underlying terms in the diffusion process, its sensitivity to the temporal discretization scheme, its inherent inaccuracy due to the intractability of the conditional score function. We also show that the diffusion posterior sampler can be seen as a specific case of

---

[2]`https://github.com/LeviBorodenko/motionblur`

[3]For Gaussian blur and random inpainting, Flash-Diffusion uses randomly sampled, but less severe degradation operators than our experimental setup.

our optimal control-based sampler. Finally, leveraging the improvements granted by our solver, we validate the performance of our algorithm on several inverse problem tasks across several datasets, and demonstrate highly competitive results.

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

# A Impact Statement

This paper builds on a large body of existing work and presents an improved technique for solving generic nonlinear inverse problems, which can be seen as a generalization of guided diffusion modeling. Controlling the diffusion process in a generative model has many societal applications, and thus a broad range of downstream impacts. We believe that understanding the capabilities and limitations of such models in a public forum and open community is essential for practical and responsible integration of these technologies with society. However, the ideas presented in this work, as well as any other work in this field, must be deployed with caution to the inherent dangers of these technologies.

# B Deriving the Iterative Linear Quadratic Regulator (iLQR)

Differential Dynamic Programming (DDP) is a very popular trajectory optimization algorithm that has a rich history of theoretical results Jacobson [1968] as well as successful practical applications in robotics Tassa et al. [2007, 2014], aerospace Houghton et al. [2022], Sasaki et al. [2022] and biomechanics Todorov and Li [2005]. It falls under the class of *indirect methods* for trajectory optimization, wherein Bellman's principle of optimality defines the so-called optimal value-function which in turn can be used to determine the optimal control. This is in contrast to so-called *direct methods* which cast the problem at hand into a nonlinear constrained optimization problem.

To formulate an optimal control algorithm we first define the state transition function of a dynamical system as

$$\mathbf{x}_{t-1} = \mathbf{h}(\mathbf{x}_t, \mathbf{u}_t). \tag{33}$$

The next ingredient that we need for our optimal control approach is a cost function $J(\mathbf{x}_t, \mathbf{u}_t) \in \mathbb{R}$. This is used to define a performance criterion that iLQR can optimize with respect to the set of controls $\{\mathbf{u}_t\}_{t=T}^{t=1}$ (i.e., the control trajectory going backwards from time $t = T$ to $t = 1$). The cost-function is defined as follows:

$$J_T = \sum_{t=T}^{1} \ell_t(\mathbf{x}_t, \mathbf{u}_t) + \ell_0(\mathbf{x}_0), \tag{34}$$

where, $\ell_t$ and $\ell_0$ are scalar-valued functions which are commonly referred to as the *running* cost-function and the *terminal* cost-function respectively.

To obtain the sequence of optimal controls, we employ the dynamic programming principle. To do so, we first introduce the notion of the Value-function defined as follows:

$$V(\mathbf{x}_t, t) = \min_{\{\mathbf{u}_n\}_{n=t}^{n=1}} J_t = \min_{\{\mathbf{u}_n\}_{n=t}^{n=1}} \Big[ \sum_{n=t}^{1} \ell_n(\mathbf{x}_n, \mathbf{u}_n) + \ell_0(\mathbf{x}_0) \Big] \tag{35}$$

Intuitively, the Value-function resembles the *optimal cost-to-go* starting from time step $t$ and state $\mathbf{x}_t$ until the end of the time horizon (i.e., $t = 0$). Using this definition, one can easily derive the following recursive relation also known as *Bellman's Principle of Optimality*:

$$V(\mathbf{x}_t, t) = \min_{\mathbf{u}_t} \Big[ \ell_t(\mathbf{x}_t, \mathbf{u}_t) + V(\mathbf{x}_{t-1}, t-1) \Big]. \tag{36}$$

A often useful defintion used in the derivation of the iLQR Riccati equations is that of the State-Action Value-Function $Q(\mathbf{x}_t, \mathbf{u}_t)$ given by,

$$Q(\mathbf{x}_t, \mathbf{u}_t) = \ell_t(\mathbf{x}_t, \mathbf{u}_t) + V(\mathbf{x}_{t-1}, t-1) \tag{37}$$

$$\text{Therefore, } V(\mathbf{x}_t, t) = \min_{\mathbf{u}_t} Q(\mathbf{x}_t, \mathbf{u}_t) \tag{38}$$

A sketch of the derivation of the Riccati equations is as follows: we take second-order Taylor expansions of both $Q(\mathbf{x}_t, \mathbf{u}_t)$ and $V(\mathbf{x}_t, t)$ around *nominal* state and action trajectories of $\{\bar{\mathbf{x}}_t\}_{t=T}^{t=0}$ and $\{\bar{\mathbf{u}}_t\}_{t=T}^{t=1}$ respectively. Next, we substitute these into Eq.(37) and equate the first- and second-

order terms to yield the following relations between the derivatives of $Q$, $\ell$ and $V$:

$$Q_{\mathbf{x}} = \ell_{\mathbf{x}} + \mathbf{h}_{\mathbf{x}}^T V_{\mathbf{x}}' \tag{39}$$

$$Q_{\mathbf{u}} = \ell_{\mathbf{u}} + \mathbf{h}_{\mathbf{u}}^T V_{\mathbf{x}}' \tag{40}$$

$$Q_{\mathbf{xx}} = \ell_{\mathbf{xx}} + \mathbf{h}_{\mathbf{x}}^T V_{\mathbf{xx}}' \mathbf{h}_{\mathbf{x}} \tag{41}$$

$$Q_{\mathbf{xu}} = \ell_{\mathbf{xu}} + \mathbf{h}_{\mathbf{x}}^T V_{\mathbf{xx}}' \mathbf{h}_{\mathbf{u}} \tag{42}$$

$$Q_{\mathbf{ux}} = \ell_{\mathbf{ux}} + \mathbf{h}_{\mathbf{u}}^T V_{\mathbf{xx}}' \mathbf{h}_{\mathbf{x}} \tag{43}$$

$$Q_{\mathbf{uu}} = \ell_{\mathbf{uu}} + \mathbf{h}_{\mathbf{u}}^T V_{\mathbf{xx}}' \mathbf{h}_{\mathbf{u}}, \tag{44}$$

where $\mathbf{h}_{\mathbf{x}_t}$ and $\mathbf{h}_{\mathbf{u}_t}$ are the Jacobians of the dynamics function $\mathbf{h}(\mathbf{x}_t, \mathbf{u}_t)$, evaluated at time step $t$, w.r.t the state and the control vectors respectively. For ease of notation, we have dropped the subscript $t$ and therefore all derivatives above should be considered to be evaluated at time step $t$, while we use $V_{\mathbf{x}}'$ and $V_{\mathbf{xx}}'$ above to indicate the gradient and hessian of the Value-function evaluated at the next time step (i.e., at time step $t - 1$).

Next, we substitute for the second-order approximation of $Q(\mathbf{x}_t, \mathbf{u}_t)$ into Eq. (38) and note that $\mathbf{u}_t$ can be written in terms of the nominal control as follows:

$$\mathbf{u}_t = \bar{\mathbf{u}}_t + \delta\mathbf{u}_t.$$

This results in a quadratic objective w.r.t $\delta\mathbf{u}_t$ and the minimization in Eq. (38) can be performed exactly resulting in the following optimal perturbation from the nominal control trajectory:

$$\delta\mathbf{u}_t^* = \mathbf{k}_t + \mathbf{K}_t \delta\mathbf{x}_t \tag{45}$$

where, the feedforward and feedback gains are given by the following expressions:

$$\mathbf{k} = -Q_{\mathbf{uu}}^{-1} Q_{\mathbf{u}} \tag{46}$$

$$\mathbf{K} = -Q_{\mathbf{uu}}^{-1} Q_{\mathbf{ux}} \tag{47}$$

Finally, by substituting for the optimal $\delta\mathbf{u}_t^*$ back into Eq.(38), we can drop the $\min$ operator and equate the first- and second-order terms on both sides. This results the following Riccati equations:

$$V_{\mathbf{x}} = Q_{\mathbf{x}} - \mathbf{K}^T Q_{\mathbf{uu}} \mathbf{k} \tag{48}$$

$$V_{\mathbf{xx}} = Q_{\mathbf{xx}} - \mathbf{K}^T Q_{\mathbf{uu}} \mathbf{K}. \tag{49}$$

This concludes the sketch derivation of the Riccati equations. The algorithm roughly proceeds as follows:

1. We start with an initial guess of the the nominal control trajectory $\{\bar{\mathbf{u}}_t\}_{t=T}^1$ and generate the corresponding nominal state trajectory $\{\bar{\mathbf{x}}_t\}_{t=T}^0$ using $\mathbf{x}_t = \mathbf{h}(\mathbf{x}_{t+1}, \mathbf{u}_{t+1})$.

2. By noticing from Eq. (35) that $V(\mathbf{x}_0, 0) = \ell(\mathbf{x}_0)$ we can obtain expressions for $V_{\mathbf{x}}$ and $V_{\mathbf{xx}}$ evaluated at $\bar{\mathbf{x}}_0$.

3. Next, we compute the derivatives of $Q$ given by equations. (39)-(44) using $\{\bar{\mathbf{u}}_t\}_{t=T}^1$ and $\{\bar{\mathbf{x}}_t\}_{t=T}^1$.

4. Using the derivatives of $Q$, we can compute the feedforward and feedback gains using equations (46)-(47).

5. Finally, using the Riccati equations (48)-(49), we can propagate both $V_{\mathbf{x}}$ and $V_{\mathbf{xx}}$ one step backwards in time.

6. We then repeat the steps 3, 4 and 5 until we backpropagate the derivatives of $V$ to time step $t = T$.

7. This completes one iteration of iLQR. At the end of each iteration the gains are used to produce the updated nominal control trajectory as follows:

$$\bar{\mathbf{u}}_t^* = \bar{\mathbf{u}}_t + \alpha\mathbf{k} + \mathbf{K}(\bar{\mathbf{x}}_t - \mathbf{x}_t) \tag{50}$$

where, $\mathbf{x}_t$ is the state obtained by unrolling the dynamics subject to the updated controls:

$$\mathbf{x}_t = \mathbf{h}(\mathbf{x}_{t+1}, \bar{\mathbf{u}}_{t+1}^*).$$

8. The new nominal control trajectory $\bar{\mathbf{u}}_t^*$ is used to produce a new nominal state trajectory $\bar{\mathbf{x}}_t^*$ and the algorithm is repeated from step 2 onwards until convergence or a fixed number of iterations.

## C  Proofs

**Theorem 4.1.** *Let Eq. 3 be the discretized sampling equation for the diffusion model with **output perturbation mode** control (Eq. 18). Moreover, let the terminal cost*

$$\ell_0(\mathbf{x}_0) = -\log p(\mathbf{y}|\mathbf{x}_0) \tag{27}$$

*be twice-differentiable and the running costs*

$$\ell_t(\mathbf{x}_t, \mathbf{u}_t) = 0. \tag{28}$$

*Then the iterative linear quadratic regulator with Tikhonov regularizer $\alpha$ produces the control*

$$\mathbf{u}_t = \alpha \nabla_{\mathbf{x}_t} \log p(\mathbf{y}|\mathbf{x}_0). \tag{29}$$

*Proof.* We demonstrate the result via induction for $t = 1, \dots, T$.

Since we assume that $\ell_{\mathbf{uu}} = \mathbf{0}$, $V_{\mathbf{xx}}$ vanishes:

$$V_{\mathbf{xx}} = Q_{\mathbf{xx}} - Q_{\mathbf{ux}}^T Q_{\mathbf{uu}}^{-1} Q_{\mathbf{ux}} \tag{51}$$

$$= h_{\mathbf{x}}^T V_{\mathbf{xx}}' h_{\mathbf{x}} - h_{\mathbf{x}}^T V_{\mathbf{xx}}' (V_{\mathbf{xx}}')^{-1} V_{\mathbf{xx}}' h_{\mathbf{x}} \tag{52}$$

$$= \mathbf{0}. \tag{53}$$

Similarly, $V_{\mathbf{x}}$ also greatly simplifies as

$$V_{\mathbf{x}} = Q_{\mathbf{x}} + Q_{\mathbf{ux}}^T Q_{\mathbf{uu}}^{-1} Q_{\mathbf{u}} \tag{54}$$

$$= h_{\mathbf{x}}^T V_{\mathbf{x}}' + h_{\mathbf{x}}^T V_{\mathbf{xx}}' (V_{\mathbf{xx}}')^{-1} V_{\mathbf{x}}' \tag{55}$$

$$= h_{\mathbf{x}}^T V_{\mathbf{x}}'. \tag{56}$$

Turning to the Tikhonov regularized feedforward term,

$$\mathbf{k} = -Q_{\mathbf{uu}}^{-1} Q_{\mathbf{u}} \tag{57}$$

$$= -(h_{\mathbf{x}}^T \underbrace{V_{\mathbf{xx}}}_{\mathbf{0}} h_{\mathbf{x}} + \alpha \mathbf{I})^{-1} Q_{\mathbf{u}} \tag{58}$$

$$= -(\mathbf{0} + \alpha \mathbf{I})^{-1} Q_{\mathbf{u}} \tag{59}$$

$$= -\frac{1}{\alpha} V_{\mathbf{x}}'. \tag{60}$$

Finally, the feedback term disappears due to the vanishing $V_{\mathbf{xx}}$

$$\mathbf{K} = -Q_{\mathbf{uu}}^{-1} Q_{\mathbf{ux}} \tag{61}$$

$$= \mathbf{0}. \tag{62}$$

Explicitly denoting the dependence of $V_{\mathbf{x}}$ and $V_{\mathbf{x}}'$ on $t$, we can rewrite Eq. 56 as

$$V_{\mathbf{x}}^{(t)} = h_{\mathbf{x}}^T V_{\mathbf{x}}^{(t-1)}$$

$$= \frac{\partial \mathbf{x}_{t-1}}{\partial \mathbf{x}_t} \frac{\partial}{\partial \mathbf{x}_{t-1}} V.$$

Combining this observation with the fact that $\ell_0 = -\log p(\mathbf{y}|\mathbf{x}_0)$, we can conclude that

$$V_{\mathbf{x}}^{(t)} = -\nabla_{\mathbf{x}_t} \log p(\mathbf{y}|\mathbf{x}_0), \tag{63}$$

where $\mathbf{x}_0$ depends on $\mathbf{x}_t$ via the state transition function $\mathbf{h}$ (Eq. 18). Therefore, we have that

$$\mathbf{k} = -\frac{1}{\alpha} V_{\mathbf{x}}'$$

$$= \frac{1}{\alpha} \nabla_{\mathbf{x}_t} \log p(\mathbf{y}|\mathbf{x}_0)$$

$$\mathbf{K} = 0.$$

Finally, given our action update (Eq. 15), we can conclude our desired result

$$\mathbf{u}_t = \frac{1}{\alpha} \nabla_{\mathbf{x}_t} \log p(\mathbf{y}|\mathbf{x}_0). \tag{64}$$

$$\square$$

**Lemma C.1.** *Under the deterministic sampler with **output perturbation mode** control, $\alpha = \frac{1}{g(t)^2 \Delta t}$ recovers posterior sampling (Eq. 9).*

*Proof.* Substituting in $\alpha = \frac{1}{g(t)^2 \Delta t}$ to Eq. 29, we observe that Eq. 18 can now be written as

$$\mathbf{x}_{t-1} = [f(\mathbf{x}_t) - \frac{1}{2}g(t)^2(\nabla_{\mathbf{x}_t} \log p_t(\mathbf{x}_t) + \nabla_{\mathbf{x}_t} \log p_t(\mathbf{y}|\mathbf{x}_0))]\Delta t. \tag{65}$$

Under the determinstic sampler, we can conclude that $\log p_t(\mathbf{y}|\mathbf{x}_0) = \log p_t(\mathbf{y}|\mathbf{x}_t)$, since each $\mathbf{x}_t$ has a *unique* path through the sample space. Therefore, we conclude that Eq. 65 resembles the ideal posterior sampler equation 9. We conclude our proof. $\square$

**Theorem 4.3.** *Let Eq. 3 be the discretized sampling equation for the diffusion model with **input perturbation mode** control (Eq. 17). Moreover, let*

$$\ell_0(\mathbf{x}_0) = \log p(\mathbf{y}|\mathbf{x}_0), \tag{30}$$

*and the running costs*

$$\ell_t(\mathbf{x}_t, \mathbf{u}_t) = 0. \tag{31}$$

*Then the iterative linear quadratic regulator with Tikhonov regularizer $\alpha = \frac{1}{g(t)^2 \Delta t}$ produces the dynamical sytem*

$$\widetilde{\mathbf{x}}_t = \widetilde{\mathbf{x}}_t + [f(\widetilde{\mathbf{x}}_t) - \frac{1}{2}g(t)^2(\nabla_{\mathbf{x}} \log p_t(\widetilde{\mathbf{x}}_t)$$
$$+ \nabla_{\mathbf{x}} \log p_t(\mathbf{y}|\mathbf{x}_t))]\Delta t, \tag{32}$$

*where $\widetilde{\mathbf{x}}_t := \mathbf{x}_t + \mathbf{u}_t$.*

*Proof.* We similarly demonstrate the result via induction for $t = 1, \ldots, T$.

Again, assuming that $\ell_{\mathbf{uu}} = 0$, $V_{\mathbf{xx}}$ vanishes:

$$V_{\mathbf{xx}} = Q_{\mathbf{xx}} - Q_{\mathbf{ux}}^T Q_{\mathbf{uu}}^{-1} Q_{\mathbf{ux}} \tag{66}$$
$$= Q_{\mathbf{xx}} - Q_{\mathbf{xx}}(\underbrace{\ell_{\mathbf{uu}}}_{=\mathbf{0}} + Q_{\mathbf{xx}})^{-1} Q_{\mathbf{xx}} \tag{67}$$
$$= \mathbf{0}, \tag{68}$$

whereas $V_{\mathbf{x}}$ greatly simplifies as

$$V_{\mathbf{x}} = Q_{\mathbf{x}} + Q_{\mathbf{ux}}^T Q_{\mathbf{uu}}^{-1} Q_{\mathbf{u}} \tag{69}$$
$$= h_{\mathbf{x}}^T V_{\mathbf{x}}'. \tag{70}$$

Turning to the feedforward and feedback terms, we have

$$\mathbf{k} = -Q_{\mathbf{uu}}^{-1} Q_{\mathbf{u}} \tag{71}$$
$$= -(h_{\mathbf{x}}^T \underbrace{V_{\mathbf{xx}}}_{\mathbf{0}} h_{\mathbf{x}} + \alpha \mathbf{I})^{-1} Q_{\mathbf{u}} \tag{72}$$
$$= -(\mathbf{0} + \alpha \mathbf{I})^{-1} Q_{\mathbf{u}} \tag{73}$$
$$= -\frac{1}{\alpha} h_{\mathbf{x}}^T V_{\mathbf{x}}', \tag{74}$$

and

$$\mathbf{K} = -Q_{\mathbf{uu}}^{-1} Q_{\mathbf{ux}} \tag{75}$$
$$= \mathbf{0}. \tag{76}$$

We observe that

$$V_{\mathbf{x}}^{(t)} = -\frac{1}{\alpha} h_{\mathbf{x}}^T V_{\mathbf{x}}^{(t-1)}.$$

|  | SR ×4 | Random Inpainting | Box Inpainting | Gaussian Deblurring | Motion Deblurring |
|---|---|---|---|---|---|
| `T` | 50 | 50 | 50 | 50 | 50 |
| `num_iters` | 50 | 100 | 100 | 100 | 100 |
| `step_size` | $1e-3$ | $1e-3$ | $1e-3$ | $1e-3$ | $1e-3$ |
| $\ell_0(\mathbf{x}_0)$ | $\|\mathcal{A}(\mathbf{x}_0)-\mathbf{y}\|$ | $\|\mathcal{A}(\mathbf{x}_0)-\mathbf{y}\|$ | $\|\mathcal{A}(\mathbf{x}_0)-\mathbf{y}\|$ | $\|\mathcal{A}(\mathbf{x}_0)-\mathbf{y}\|$ | $\|\mathcal{A}(\mathbf{x}_0)-\mathbf{y}\|$ |
| $\alpha$ | $1e-4$ | $1e-4$ | $1e-4$ | $1e-4$ | $1e-4$ |
| $\ell_t(\mathbf{x}_t,\mathbf{u}_t)$ | $\alpha\|\mathbf{u}_t\|$ | $\alpha\|\mathbf{u}_t\|$ | $\alpha\|\mathbf{u}_t\|$ | $\alpha\|\mathbf{u}_t\|$ | $\alpha\|\mathbf{u}_t\|$ |
| $k$ | 1 | 1 | 1 | 1 | 1 |
| `control_mode` | input mode | input mode | input mode | input mode | input mode |

Table 2: Hyperparameters for FFHQ experiments.

Therefore, noting that $V_\mathbf{x}^{(0)} = \log p(\mathbf{y}|\mathbf{x}_0)$, we have

$$\mathbf{k} = -V_\mathbf{x}^{(t)}$$
$$= -\frac{1}{\alpha}(h_\mathbf{x}^{(t)})^T V_\mathbf{x}^{(t-1)}$$
$$= -\frac{1}{\alpha}\nabla_{\mathbf{x}_t}\log p(\mathbf{y}|\mathbf{x}_0)$$
$$= -\frac{1}{\alpha}\nabla_{\mathbf{x}_t}\log p(\mathbf{y}|\mathbf{x}_0(\mathbf{x}_t)).$$

Applying the feedforward terms to the diffusion sampling process, we have

$$\mathbf{x}_{t-1} = (\mathbf{x}_t + \mathbf{u}_t) + [f(\mathbf{x}_t + \mathbf{u}_t)$$
$$- \frac{1}{2}g(t)^2\nabla_\mathbf{x}\log p_t(\mathbf{x}_t + \mathbf{u}_t)]\Delta t.$$

We define the intermediary variable

$$\widetilde{\mathbf{x}}_t = \mathbf{x}_t + \mathbf{u}_t, \tag{77}$$

which has dynamics

$$\widetilde{\mathbf{x}}_t = \widetilde{\mathbf{x}}_t + [f(\widetilde{\mathbf{x}}_t) - \frac{1}{2}g(t)^2\nabla_\mathbf{x}\log p_t(\widetilde{\mathbf{x}}_t)]\Delta t + \mathbf{u}_t. \tag{78}$$

We now can see that, letting $\alpha = \Delta t g(t)^2$, we obtain

$$\widetilde{\mathbf{x}}_t = \widetilde{\mathbf{x}}_t + [f(\widetilde{\mathbf{x}}_t) - \frac{1}{2}g(t)^2(\nabla_\mathbf{x}\log p_t(\widetilde{\mathbf{x}}_t) + \nabla_\mathbf{x}\log p_t(\mathbf{y}|\mathbf{x}_0))]\Delta t.$$

. □

# D Implementation

For all experiments, we use publicly available datasets and pre-trained model weights. For the FFHQ $256 \times 256$ experiments, we use the last 1K images of the dataset for evaluation. For MNIST, we do not use images directly in the inverse classification task. The images were only used for training the pretrained diffusion model.

For models, we used the pretrained weights from Chung et al. [2023a] for FFHQ $256 \times 256$ tasks, and the Hugging Face `1aurent/mnist-28` diffusion model for MNIST experiments. No further training is performed on any models. Further hyperparameters can be found in Table 2. For the classifier $p(\mathbf{y}|\mathbf{x})$ in MNIST class-guided classification, we use a simple convolutional neural network with two convolutional layers and two MLP layers, trained on the entire MNIST dataset.

## D.1 High Dimensional Control

To speed up our proposed method, we leverage the following three modifications to the standard iLQR algorithm.

|  | SR ×4 | | | Random Inpainting | | | Box Inpainting | | | Gaussian Deblurring | | | Motion Deblurring | | |
|---|---|---|---|---|---|---|---|---|---|---|---|---|---|---|---|
|  | PSNR ↑ | SSIM ↑ | MSE ↓ | PSNR ↑ | SSIM ↑ | MSE ↓ | PSNR ↑ | SSIM ↑ | MSE ↓ | PSNR ↑ | SSIM ↑ | MSE ↓ | PSNR ↑ | SSIM ↑ | MSE ↓ |
| Ours (T = 50) | **27.45** | 0.792 | **117.0** | **31.84** | **0.882** | **42.57** | 25.33 | 0.804 | 190.6 | **24.99** | 0.694 | 206.1 | 25.08 | 0.721 | **201.9** |
| DPS (T = 1000) | 25.67 | 0.852 | 176.2 | 22.47 | 0.873 | 368.2 | 25.23 | **0.851** | 195.0 | 24.25 | **0.811** | 244.4 | 24.92 | **0.859** | 209.4 |
| DDRM (T = 1000) | 25.36 | 0.835 | 189.3 | 22.24 | 0.869 | 388.2 | 9.19 | 0.319 | 7835 | 23.36 | 0.767 | 300.0 | - | - | - |
| MCG (T = 1000) | 20.05 | 0.559 | 642.8 | 19.97 | 0.703 | 654.8 | 21.57 | 0.751 | 453.0 | 6.72 | 0.051 | 13838 | 6.72 | 0.055 | 13838 |
| PNP-ADMM | 26.55 | **0.865** | 143.9 | 11.65 | 0.642 | 4447 | 8.41 | 0.325 | 9377 | 24.93 | 0.812 | 208.9 | 24.65 | 0.825 | 222.9 |
| Score-SDE (T = 1000) | 17.62 | 0.617 | 1124 | 18.51 | 0.678 | 916.4 | 13.52 | 0.437 | 2891 | 7.12 | 0.109 | 12620 | 6.58 | 0.102 | 14291 |
| ADMM-TV | 23.86 | 0.803 | 267.4 | 17.81 | 0.814 | 1076 | 22.03 | 0.784 | 407.5 | 22.37 | 0.801 | 376.8 | 21.36 | 0.758 | 475.4 |

Table 3: Quantitative evaluation (PSNR, SSIM, MSE) of performance on inverse problems on the FFHQ 256x256-1K dataset.

|  | SR ×4 | | | | Random Inpainting | | | |
|---|---|---|---|---|---|---|---|---|
|  | LPIPS ↓ | PSNR ↑ | SSIM ↑ | MSE ↓ | LPIPS ↓ | PSNR ↑ | SSIM ↑ | MSE ↓ |
| $k = 0$ | 0.254 | 24.00 | 0.691 | 141.2 | 0.121 | 28.33 | 0.755 | 56.74 |
| $k = 1$ | 0.171 | 27.45 | 0.792 | 117.0 | 0.053 | 31.84 | 0.882 | 42.57 |
| $k = 4$ | 0.171 | 27.47 | 0.794 | 116.4 | 0.052 | 31.99 | 0.883 | 41.12 |
| $k = 16$ | 0.170 | 27.43 | 0.799 | 117.5 | 0.050 | 32.12 | 0.891 | 39.90 |

Table 4: Ablative study on the effect of *rank* in the low rank and matrix-free approximations on performance (LPIPS, PSNR, SSIM, NMSE) of our proposed model on the FFHQ 256x256-1K dataset dataset.

**Randomized Low-Rank Approximation** The first and second order terms in Eqs. (19-25) are corresponding Taylor expansions of deep neural functions. Even with the use of automatic differentiation libraries, the formation of these matrices is incredibly expensive, requiring at least $\dim(\mathbf{x})$ backpropagation passes (where $\dim(\mathbf{x}) \approx 39B$ in some experiments). To reduce the cost of computing these matrices, we utilize their known low rank structure Sagun et al. [2017], Oymak et al. [2019].

Leveraging advanced techniques in randomized numerical linear algebra, we estimate Eqs. (19-25) using randomized SVD Halko et al. [2011]. For any matrix $\mathbf{A} \in \mathbb{R}^{m \times n}$ this is a four step process. 1) We sample a random matrix $\mathbf{\Omega} \sim \mathcal{N}(\mathbf{0}, \mathbf{I}_{n \times k})$. 2) We obtain $\mathbf{A\Omega} = \mathbf{Y} \in \mathbb{R}^{m \times k}$. 3) We form a basis over the columns of $\mathbf{Y}$, e.g. by taking the $\mathbf{Q}$ matrix in a $\mathbf{QR}$ factorization $\mathbf{QR} = \mathbf{Y}$. 4) We approximate $\mathbf{A} \approx \mathbf{Q}^T \mathbf{QA}$.

Notably, we observe that when $\mathbf{A}$ is a Jacobian (or Hessian) matrix, it can be approximated purely through Jacobian-vector and vector-Jacobian (Hessian-vector and vector-Hessian, resp.) products — *without ever materializing $\mathbf{A}$ itself*. Moreover, a key result in randomized linear algebra is that this algorithm can approximate $\mathbf{A}$ up to accuracy $\mathcal{O}(mnk\sigma_{k+1})$ (Theorem 1.1 in Halko et al. [2011]). Notably, if $\mathbf{A}$ has low rank structure where $\exists k$ such that the $k + 1$th singular value $\sigma_{k+1} = 0$, then the approximation is exact.

**Matrix-Free Evaluation** Inspired by matrix-free techniques in numerical optimization Knoll and Keyes [2004], we demonstrate a strategy for forming the action update (15) without materializing the costly $\dim(\mathbf{x}) \times \dim(\mathbf{x})$ matrices in the iLQR algorithm (19-25), which we shall denote as an

|  | SR ×4 | | | | Random Inpainting | | | |
|---|---|---|---|---|---|---|---|---|
|  | LPIPS ↓ | PSNR ↑ | SSIM ↑ | MSE ↓ | LPIPS ↓ | PSNR ↑ | SSIM ↑ | MSE ↓ |
| $\alpha = 0$ | - | - | - | - | - | - | - | - |
| $\alpha = 1e-7$ | 0.173 | 27.49 | 0.794 | 115.9 | 0.050 | 31.80 | 0.879 | 42.96 |
| $\alpha = 1e-4$ | 0.171 | 27.45 | 0.792 | 117.0 | 0.053 | 31.84 | 0.882 | 42.57 |
| $\alpha = 1$ | 0.172 | 27.43 | 0.799 | 117.5 | 0.050 | 31.85 | 0.891 | 42.47 |
| $\alpha$ from Lemma 4.2 | 0.170 | 27.44 | 0.788 | 117.3 | 0.051 | 31.86 | 0.880 | 42.44 |

Table 5: Ablative study on the effect of the Tikhonov regularization coefficient $\alpha$ on performance (LPIPS, PSNR, SSIM, NMSE) of our proposed model on the FFHQ 256x256-1K dataset dataset. No results are reported for $\alpha = 0$, as the algorithm encountered numerical precision errors during matrix inversion.

| | SR ×4 | | | | Random Inpainting | | | |
|---|---|---|---|---|---|---|---|---|
| | LPIPS ↓ | PSNR ↑ | SSIM ↑ | MSE ↓ | LPIPS ↓ | PSNR ↑ | SSIM ↑ | MSE ↓ |
| $T = 10$ | 0.198 | 27.48 | 0.783 | 125.6 | 0.168 | 27.46 | 0.771 | 123.7 |
| $T = 20$ | 0.1923 | 31.79 | 0.859 | 117.0 | 0.108 | 34.41 | 0.910 | 42.57 |
| $T = 50$ | 0.171 | 27.45 | 0.792 | 90.79 | 0.053 | 31.84 | 0.882 | 40.56 |
| $T = 200$ | 0.155 | 28.55 | 0.811 | 43.05 | 0.048 | 32.05 | 0.899 | 23.17 |

Table 6: Ablative study on the effect of $T$ on performance (LPIPS, PSNR, SSIM, NMSE) of our proposed model on the FFHQ 256x256-1K dataset dataset.

indexed set of matrices $\{\mathbf{A}_i\}$. We do this by forming projections of each $\mathbf{A}_i$ against a corresponding set of $\dim(\mathbf{x}) \times \ell$ column-orthogonal matrices $\{\mathbf{Q}_i\}$, which we denote as $\mathbf{B}_i := \mathbf{Q}_i^T \mathbf{A}_i$. These matrices can then be stored at reduced cost as $(\mathbf{Q}_i, \mathbf{B}_i)$ pairs.

Matrix multiplications between any $\mathbf{A}_i \mathbf{A}_j$ can then be approximated up to rank $\ell$ with respect to the projected matrix, $\mathbf{Q}_i \mathbf{A}_{i,\mathbf{Q}_i}$, i.e.

$$\mathbf{A}_i \mathbf{A}_j \approx \mathbf{Q}_i \mathbf{B}_i \mathbf{Q}_j^T \mathbf{B}_j. \tag{79}$$

However, to prevent materialization of the full size of any matrices, we drop the leading $\mathbf{Q}_i$, obtaining a new projected-matrix pair $(\mathbf{Q_k}, \mathbf{B_k})$, where $\mathbf{Q_k} = \mathbf{Q}_i$.

**Adam Optimizer** Finally, we precondition gradients via the Adam optimizer Kingma and Ba [2014] before applying the feedback gains, rather than applying a backtracking line search Tassa et al. [2014], resulting in the action update

$$\mathbf{u}_t = \mathbf{P}\mathbf{k}_t + \mathbf{K}_t(\mathbf{x}_t - \mathbf{x}'_t), \tag{80}$$

where $\mathbf{P}$ is the preconditioning matrix produced by the Adam optimizer. This reduces the overall runtime of the algorithm while still accounting for second-order information that respects the nonlinearity of the optimization landscape.

### D.2 Computational Complexity Analysis

Incorporating all three modifications, we can provide a realistic runtime and space complexity analysis of our presented algorithm with respect to the rank $k$, the data dimension $d$, diffusion steps $m$, and number of iLQR iterations $n$.

Combining both the low rank and matrix-free approximations, we obtain the updated equations for input mode perturbation (where projection matrices are written as $\mathbf{P}$ to avoid overloading the $Q$ function notation):

$$Q_\mathbf{x} = \mathbf{h}_\mathbf{x}^T V'_\mathbf{x} \tag{81}$$

$$Q_\mathbf{u} = \ell_\mathbf{u} + \mathbf{h}_\mathbf{x}^T V'_\mathbf{x} \tag{82}$$

$$\mathbf{P}Q_{\mathbf{xx}}\mathbf{P}^T = \mathbf{P}Q_{\mathbf{ux}}\mathbf{P}^T = \mathbf{P}Q_{\mathbf{xu}}\mathbf{P}^T = \mathbf{P}\mathbf{h}_\mathbf{x}^T V'_{\mathbf{xx}}\mathbf{h}_\mathbf{x}\mathbf{P}^T \tag{83}$$

$$\mathbf{P}Q_{\mathbf{uu}}\mathbf{P}^T = \mathbf{P}\ell_{\mathbf{uu}}\mathbf{P}^T + \mathbf{P}\mathbf{h}_\mathbf{x}^T V'_{\mathbf{xx}}\mathbf{h}_\mathbf{x}\mathbf{P}^T. \tag{84}$$

To simplify notation, each projection matrix $\mathbf{P}$ is the same — in reality, this need not be the case. Note that $\mathbf{Q}_x$ and $\mathbf{Q}_u$ are simply of size $d$ and therefore image-sized. For all our datasets, these each take 0.2 MB to store and are therefore negligible, and we do not project these variables. When $\ell_{\mathbf{uu}}$ is diagonal (as it is in our case), we can obtain the projected inverse for $Q_{\mathbf{uu}}$ as

$$\mathbf{P}Q_{\mathbf{uu}}^{-1}\mathbf{P}^T = \mathbf{P}\ell_{\mathbf{uu}}^{-1}\mathbf{P}^T + \mathbf{P}\ell_{\mathbf{uu}}^{-1}\mathbf{P}^T(\mathbf{C}^{-1} + \mathbf{P}^T\ell_{\mathbf{uu}}^{-1}\mathbf{P})^{-1}\mathbf{P}\ell_{\mathbf{uu}}^{-1}\mathbf{P}^T \quad \text{where } \mathbf{C} = \mathbf{P}\mathbf{h}_\mathbf{x}^T V'_{\mathbf{xx}}\mathbf{h}_\mathbf{x}\mathbf{P} \tag{85}$$

via a direct application of the Woodbury matrix inversion formula Petersen et al. [2008], which has cost $\mathcal{O}(k^3 + kd^2)$. Finally, we compute the projected updates $V_{\mathbf{xx}}, \mathbf{K}$ as well as the full-precision

$V_{\mathbf{x}}, \mathbf{k}$ terms via

$$\mathbf{k} = -\mathbf{P}^T\mathbf{P}Q_{\mathbf{uu}}^{-1}\mathbf{P}^T\mathbf{P}Q_{\mathbf{u}} \tag{86}$$

$$V_{\mathbf{x}} = Q_{\mathbf{x}} - \mathbf{P}^T\mathbf{PK}^T\mathbf{P}^T\mathbf{P}Q_{\mathbf{uu}}\mathbf{P}^T\mathbf{Pk} \tag{87}$$

$$\mathbf{PKP}^T = -\mathbf{P}Q_{\mathbf{uu}}^{-1}\mathbf{P}^T\mathbf{P}Q_{\mathbf{ux}}\mathbf{P}^T \tag{88}$$

$$\mathbf{P}V_{\mathbf{xx}}\mathbf{P}^T = \mathbf{P}Q_{\mathbf{xx}}\mathbf{P}^T - \mathbf{PK}^T\mathbf{P}^T\mathbf{P}Q_{\mathbf{uu}}\mathbf{P}^T\mathbf{PKP}^T. \tag{89}$$

Where applicable, we leverage vector-Jacobian products from standard automatic differentiation libraries (e.g. `torch.func.vjp`) which have runtime complexity $\mathcal{O}(1)$. Computing the $V_{\mathbf{x}}, V_{\mathbf{xx}}, \mathbf{k}, \mathbf{K}$ terms in Eqs. (46)-(49) costs $\mathbf{O}(k^3 + kd^2)$ FLOPs in terms of matrix multiplications (dominated by the matrix inverse of $k \times k$ matrix $\mathbf{q}^T Q_{\mathbf{uu}}\mathbf{q}$). Crucially, it incurs $\mathcal{O}(k)$ neural function evaluations (NFEs), which dominates the runtime of the algorithm. Since this computation is performed for each diffusion step and iLQR iteration, the total runtime complexity of our algorithm is $\mathcal{O}(nm(k^3 + kd^2))$ matrix multiplication FLOPs and $\mathcal{O}(nmk)$ NFEs, with $\mathcal{O}(mk^2 + d)$ space complexity. In terms of time complexity, the NFEs are the dominating cost, accounting for $97\%$ of computation time.

## D.3 Sensitivity to Hyperparameters

In Tables 4, 5, 6, we investigate the effect of the rank of the low rank approximation and matrix-free projections, the Tikhonov regularization coefficient $\alpha$, and the diffusion time $T$ on the performance of our method on the FFHQ 256x256 dataset. We evaluate performance on the super-resolution and random inpainting tasks, with the same setup as in Section 6.

**Low-Rank and Matrix-Free Rank**   From Table 4, it is clear that there is a significant performance gain from even a rank one approximation of the first- and second-order matrices. The gains from subsequent increases in the rank approximation diminish quickly. This is because increasing the rank of the approximation only improves the approximation of the second-order terms. The first order $V_{\mathbf{x}}, Q_{\mathbf{x}}, Q_{\mathbf{u}}$ terms are always modeled exactly in $\mathcal{O}(1)$ time per iteration due to their amenability to vector-Jacobian products. From Theorems 4.1-4.3 we see that even when the second order terms are zero (i.e., the result of assumption $\ell_t = 0$), we exactly recover the true posterior sampler. Therefore, the second-order terms are less important, though still useful for imposing a quadratic trust-region regularization to the algorithm. Therefore, we ultimately choose $k = 1$ for three reasons:

1. the rank only affects the quadratic approximation of the iLQR algorithm (and does not affect our theoretical results in Theorems 4.1-4.3)

2. $k = 1$ already allows second-order propagation of the quadratic trust-region regularization, and

3. subsequent increases in $k$ have a minimal effect on the performance of the algorithm.

**Tikhonov Regularizer**   Table 5 demonstrates that our algorithm is relatively robust to the Tikhonov regularization parameter, except when $\alpha = 0$. Under this condition, any ill-conditioning of $Q_{\mathbf{uu}}$ results in division by zero errors, resulting in the failure of the algorithm. Therefore, we simply choose to let $\alpha = 1e - 4$, since the effect of Tikhonov regularizer is minimal.

**Diffusion Steps**   Finally, we observe in Table 6 that increasing the diffusion time results in higher quality samples — though at the cost of increased computation time. Therefore, choice of $T$ requires balancing computational cost and sample quality, and is ultimately highly user-dependent. When the computational and latency budget is relatively high, large $T$ can be used to improve sample quality. Conversely, when this budget is low, we find that even $T = 20$ provides reasonable samples.

