# OpenReview forum: "Solving Inverse Problems via Diffusion Optimal Control"
_NeurIPS.cc/2024/Conference — NeurIPS 2024 poster_

### Official Review · Reviewer_L2Ru · 2024-06-29

**Soundness:** 2
**Presentation:** 3
**Contribution:** 2
**Rating:** 5
**Confidence:** 3

**Summary:**

The paper addresses the limitations of existing diffusion-based inverse problem solvers, which typically frame signal recovery as a probabilistic sampling task. The authors propose a novel approach that redefines the generative process as a discrete optimal control task. Inspired by the iterative Linear Quadratic Regulator  algorithm, this new framework named diffusion optimal control, can handle various differentiable forward measurement operators, including super-resolution, inpainting, and deblurring.

**Strengths:**

1. The paper introduces a novel framework based on optimal control theory to solve diffusion-based inverse problems, moving away from the traditional probabilistic sampling approaches. This is a significant theoretical advancement.
2. The framework addresses critical drawbacks of current methods, such as the intractability of the conditional likelihood function and dependence on score network approximations. This leads to more robust and potentially more accurate solutions.

**Weaknesses:**

1. The method involves complex mathematical formulations and optimal control theory, which may pose challenges for implementation and understanding by practitioners who are not familiar with these concepts. The need to compute Jacobian and Hessian matrices, as well as the regularized inverses, may lead to significant computational demands, particularly in high-dimensional settings.

2. Lacking of enough experiments, such as MRI reconstruction or other medical images. Including more diverse datasets and additional baseline methods would provide a more comprehensive evaluation.

**Questions:**

1. What is the purpose of injecting control vectors u_t into the reverse diffusion process, and how do they influence the terminal state of the system?

2. How are the gradients V_x and Hessians V_xx of the value function used within the optimal control framework, and what is their significance?

3. Please indicate what is output of Algorithm 1.

4. Can you give some high level description about: How does using an adaptive optimizer for action updates improve the iLQR algorithm, and what impact does it have on the performance and efficiency of solving inverse problem tasks?

**Limitations:**

The method requires the forward measurement operator to be differentiable. In cases where this is not possible or practical, the applicability of the proposed framework may be limited.

The performance evaluations rely on specific pretrained models. The method’s robustness and performance with other pretrained models, or models trained on different data distributions, would need further investigation.

---

> ### Author Rebuttal · Authors · 2024-08-07
>
> We highlight our gratitude that the reviewer appreciates both the theoretical and empirical results of our work, and for bringing several insightful shortcomings of our work to our attention. Below we respond to the reviewers concerns in a point-by-point basis.
>
> **Mathematical formulations are complex and potentially challenging to implement.**
> We agree that the mathematical framework behind optimal control theory is sophisticated, and differs from that of standard diffusion models, which comes from nonequilibrium thermodynamics. However, our approach has several distinct advantages. First, we can treat the generative process as a black box dynamical system, which vastly abstracts the pre-existing mathematics in the reverse diffusion process. Practitioners may thus choose between standard diffusion modeling and optimal control-based modeling based on their expertise. Second, we can leverage extensive research from the optimal control community to improve diffusion-based inverse solvers. Third, we are able to sidestep the intractability of the conditional score function, as discussed in Section 4.
>
> **Computing Jacobian and Hessian matrices and their inverse may have significant computational demands.**
> We note that even a rank-one approximation of these matrices is sufficient, with further increases in rank providing quickly diminishing returns (see Table 4 in Appendix). Using modern randomized linear algebra libraries (e.g. randomized SVD), this brings the computation of the Hessian matrices to cost $\mathcal{O}(d)$, where $d$ is the data dimension. Therefore, the cost of our algorithm is equivalent to the cost of, e.g., performing a DPS step. (For more details, see a thorough complexity analysis in Appendix D.2.)
>
> **Including more diverse datasets would provide a more comprehensive evaluation.**
> Thank you for this insight. We agree, and additionally include results on ImageNet, and more nonlinear settings. Please see the main rebuttal and PDF.
>
> **Purpose and mechanism of injected control vectors in the reverse diffusion process.**
> The control vectors u_t are simply perturbations to the original unconditional diffusion process, and are widely used in conditional sampling with diffusion models. Analogues include the conditional score term in DPS (Chung et al., 2022), the classifier gradient in classifier guidance (Dhariwal and Nichol, 2021), or the classifier-free guidance term (Ho and Salimans, 2022).
>
> **How are $V_x$ and $V_{xx}$ used in the optimal control framework?**
> In optimal control, $V(x)$ is the value function, which intuitively represents "desirability" of each state x in the dynamical system to the user. In this case, $x$ is the reverse diffusion iterate. $V_x$ and $V_{xx}$ are simply derivatives of this value function. We can think of $V(x)$ as the negative loss of the optimal control system. Therefore we use the derivatives $V_x$ and $V_{xx}$ much like they are used in an optimization framework to guide a solver towards (locally) optimal solutions.
>
> **Please indicate the output of Algorithm 1.**
> The output of Algorithm 1 is the perturbed dynamics ${x’_t}_{t=1}^T$. The final iterate of which is the solution to the inverse problem. Thank you for this comment, we have clarified this point in the manuscript.
>
> **High level intuition on the adaptive optimizer for action updates.**
> Note that the default choice of optimizer for iLQR implementations is the backtracking line search. We instead use an adaptive optimizer, and motivate this design decision with three main observations.
>
> First, recall that given the update direction $v$ the backtracking line search finds the optimal step size $\lambda$ to update the actions by applying $u’ = u + \lambda v$ to our reverse diffusion process. Evaluating each $\lambda$ requires re-computing the inverse problem loss under the new proposed $u’$, we would have to re-run the reverse diffusion process, which is quite expensive.
>
> Second, the backtracking line search is used with generally smooth landscapes, and not as amenable to highly nonconvex settings. Since each marginal density of the diffusion process is essentially the data distribution convolved with a Gaussian distribution, it is highly nonconvex (especially at the low noise regime) and thus ill-suited for our setting.
>
> Third and last, since the diffusion process occurs on images which can be very high dimensional, our optimization space is also very high-dimensional. Adaptive optimizers such as Adam, Adagrad, RMSprop, etc. are specially designed for deep learning settings where there is a similarly high dimensional optimization space. Therefore, it is a natural choice for our problem setting.

---

> > ### Comment · Reviewer_L2Ru · 2024-08-11
> >
> > Thanks for providing detailed explanation. I changed my score to be 5.

---

### Official Review · Reviewer_u6BH · 2024-07-05

**Soundness:** 3
**Presentation:** 3
**Contribution:** 4
**Rating:** 6
**Confidence:** 3

**Summary:**

This paper proposes diffusion optimal control that solves inverse problems via posterior sampling by combining the power of a pre-trained unconditional diffusion model and the iterative Linear Quadratic Regulator algorithm to produce optimal controls that steer the reverse diffusion process to correctly recover the original signal.
The framework is general and able to handle any differentiable forward measurement operator and establishes a new baseline in image reconstruction.

**Strengths:**

* The idea of augmenting the reverse diffusion sampling process with a perturbation control is quite novel and general for arbitrary cost functions, although the paper focuses specifically on the cost for posterior sampling.
* The writing is generally good (see more comments regarding writing in questions). It is concise and to the point with good intuitions provided.
* Efforts (e.g. low-rank approximations) have been made to bring down the computational cost in iLQR for the high-dimensional image setting.
* The empirical performance of the proposed method is strong and establishes new state-of-the-art results.

**Weaknesses:**

* The runtime of the proposed method seems high and is not much discussed. iLQR is a global-in-time iterative method that could potentially require a large number of iterations to converge (and all nice things discussed in Section 4. rely on this convergence). On top of that, for each iteration, there needs to be $\Omega(T)$ matrix solves which can be quite slow given the dimension of the images (even with techniques like low-rank approximation).  It would be interesting to see ablation studies on the effect of num_iter in Algorithm 1. It would be also more convincing to report the runtime of each method in Table 1.
* There is no analysis of the approximation error of iLQR (the first and second-order Taylor approximations) in the studied setting. Specifically, it seems to me that a lot of heavy lifting is done when the control $u_t$ is designed to be only a perturbation of the reverse diffusion step. For instance, does this imply that the value function is smoother (hence the Taylor approximation is more accurate) when parameterized by $u_t$?

**Questions:**

* What is the rough runtime of each method in Table 1?
* Notations such as $p_t(x|y)$ are confusing to me. The subscript $t$ on $p$ suggests a family of distributions but I think that's not the case. My understanding of the randomness is the following. First, the random variable $x_0$ is drawn from the clean image distribution. Then $y = A(x_0) + \eta$. In parallel, we also have random variables $x_t$ obtained deterministically from $x_0$ by evolving along the ODE of the forward diffusion. A better notation in my opinion is to put the subscript $t$ on $x$, like $p(x_t|y)$.
* In (29), what is the meaning of $\nabla_{x_t} \log p(y|x_0)$?
* Line 149, what do you mean by "produces a feasible solution"? What's the meaning of being feasible?
* The texts around line 160 are hard to parse. What is the notation $p(x_0|x_t,x_0)$? What is the $x_0$-centered marginal?
* Line 176, why is $\log p(y|x_t) = \log p(y|x_0)$ an assumption, not a consequence?
* In the paragraph of Line 204, I'm confused about why the diffusion model can be taken to have randomly initialized weights. This does not seem to result in any meaningful application, since there is no information about the image distribution. For instance, in Figure 6, the produced result looks even worse than the input $y$.

Minor comments
* Line 82, what is $\theta$?
* In Section 2.3, it would be good to include the dimension of each variable. In (13), it would be good to clarify that $k, K$ all depend on $t$.
* Algorithm 1 appears unreferenced in the main text. In the input, what is $x_T$? Is it just drawn from a Gaussian? What is the output?
* Line 117, $\ell_t(x_t,u_t) = 0$, not just not depend on $x_t$. It would also be good to say here what exactly is $\ell_0$ for input/output perturbation controls instead of defining it later.
* Theorem 4.1 is missing transitions for presenting the conclusion (29).
* The values of $\alpha$ in the two theorems appear to be missing a factor of $2$.
* In (32), it should be $\widetilde{x}_{t-1}$ on the left side of the equation.
* Line 253, there is a parenthesis not closed.

**Limitations:**

There's not much discussion about the limitations in the paper. There is no potential negative societal impact.

---

> ### Author Rebuttal · Authors · 2024-08-07
>
> We greatly appreciate the reviewer’s positive assessment of theoretical contributions, for their in-depth reading of our manuscript, and for their suggested improvements. We respond to comments in detail below.
>
> **The runtime of Algorithm 1 seems high.**
> Without taking any approximations of the Hessian and Jacobian matrices in Eqs. 19-25, Algorithm 1 will indeed be relatively costly, compared to the other algorithms we compare against in Table 1. However, we maintain that our method is not significantly more expensive than competing methods. For details, see computational complexity analysis in Section D.3. Indeed, under the $T=50$ configuration, our method is slightly more expensive than competing methods. That being said, we run an equivalent runtime/flops budget in the Ours (T=20) case in Table 1, where we restrict our method to 1000 NFEs, which take up the majority (~97%) of the runtime of our algorithm. We still show significant gains over DPS, and comparable performance against the most recent state-of-the-art methods on FFHQ256-1K. Empirically, we find that our method has a similar runtime to DPS on an A6000 GPU (130s vs 125s).
>
>
> **iLQR could require a large number of iterations to converge.**
> Indeed, iLQR could take many iterations to converge. However, we observe near convergence for nearly all of the settings we consider in our work. Moreover, true convergence is not necessarily desirable, since the forward operator measurements $y$ are noisy. Therefore, a fully converged iLQR method would overfit to the noisy, and produce inferior solutions.
>
>
> **...(and all nice things discussed in Section 4. rely on this convergence).**
> We respectfully disagree. A central claim in our paper, and in Theorems 4.1-4.3 in Section 4 is the ability to compute the conditional score function $\nabla_{x_t} \log p(y | x_0)$. This claim does not rely at all on the convergence of the iLQR algorithm. In fact, this quantity is a fundamental property of the backward pass of the iLQR algorithm (the second loop in Algorithm 1). Therefore, we obtain the true $\nabla_{x_t} \log p(y | x_0)$ in every pass of our the iLQR algorithm, including the first step.
>
> **Furthermore, each iteration needs to be $\Omega(T)$ matrix solves which can be quite slow.**
> This is true. We agree with the reviewer's statement. However, the constant swallowed by the $\Omega$ notation is important, and rather small here. After an ablation study, we found (somewhat surprisingly) that even a rank-1 approximation of the relevant matrices is sufficient to obtain competitive results on our benchmarks (Table 4). Under a rank-1 approximation, many of the matrix operations are simply $\mathcal{O}(d)$, where $d$ is the size of the image.
>
> **It would be interesting to see ablation studies on the effect of num_iter in Algorithm 1.**
> This is a good point. We provide the ablation study on the FFHQ256-1K dataset with the super-resolution task, letting $T=50$ and $k=1$. Performance is evaluated with the LPIPS metric.
> | num_iter | 1 | 5 | 10 | 25 | 50 |
> |-|-|-|-|-|-|
> | | 0.491 | 0.411 | 0.322 | 0.236 | 0.171 |
>
> **No analysis of the approximation error of iLQR (the first and second-order Taylor approximations) in the studied setting.**
> We respectfully disagree. We do analyze the approximation quality of our proposed iLQR Algorithm 1 in Theorems 4.1 and 4.3. Here, we show that, in the deterministic setting, the solution $\mu_t$ is precisely the desired conditional score $\nabla \log p(y | x_0)$, which is the quantity desired in DPS-based solvers at each step. Therefore, in the deterministic setting, the approximation error arises entirely from the discretization error of the numerical solver for $x_0$ itself. In the stochastic setting, the approximation error will also come from the randomness of the diffusion process.
>
> **What is the rough runtime of each method in Table 1?**
> Below we report the approximate runtimes for each method on an NVIDIA A6000 GPU. The first two are ours with different choices of $T$.
> | (T = 50) | (T=20) | DPS | MCG | PSLD | DDNM | DDRM | Score-SDE |
> |-|-|-|-|-|-|-|-|
> | 259s | 130s | 125s | 123s | 251s | 122s | 35s | 50s|
>
> **Elaboration on $p_t(x|y)$ notation.**
> In general, $p_t(x|y)$ and $p_t(x)$ *do* come from a family of distributions, and refer to the marginal distribution of the conditional diffusion process at time $t$. Each $x_t \sim p_t(x|y)$ can be obtained, given $x_0$, via $x_t = \alpha x_0 + \sqrt{1 - \alpha} \epsilon$, where $x_0 \sim p(x | y)$ (the distribution of our solution set), and $\epsilon \sim \mathcal{N}(0, \mathbf{I})$.
>
> **In (29), what is the meaning of $\nabla_{x_t} \log⁡ 𝑝(𝑦|𝑥_0)$?**
> We apologize for the confusing notation. This is the derivative of the conditional log likelihood $\log p(y | x_0)$, which depends on $x_0$, which in the reverse diffusion process depends on $x_t$. We thank you for this question and have clarified this notation in the text immediately following (29).
>
> **What is a feasible solution in Line 149?**
> We mean feasibility in the constrained optimization sense – a solution is feasible when it satisfies all the constraints of the problem, i.e., $y \approxeq f(x)$. Since our proposed algorithm is closed loop (i.e., changes at each step $t$ depend on the final error at $t=0$), the algorithm can make global modifications to the diffusion trajectory at each $t$. To our knowledge, this is not possible for general inverse problems with competing methods like DPS, MCG, etc.

---

> ### Author Response · Authors · 2024-08-07
>
> **Line 176: Why is $\log⁡ 𝑝(𝑦|𝑥_0) = \log⁡ 𝑝(𝑦|𝑥_t)$ an assumption, not a consequence?**
> Indeed, $\log p(y | x_t) = \log p(y | x_0)$ is a consequence under the deterministic ODE dynamics. However, we also consider the setting of DPS, where the process is stochastic. Under this setting, there may be more than one $y$ (and thus $x_0$) associated with a given $x_t$, since $p(x_t|x_0)$ is Gaussian and supported everywhere. In this case, the equality is no longer guaranteed, but is assumed in many prior works (e.g., DPS).
>
> **Line 204: What is the purpose this experiment with randomly initialized weights?**
> Indeed, this application is itself not meaningful. We meant to demonstrate DPS samplers *require* a well-approximated score model to function properly, whereas we do not. Of course, the performance of our model also improves with a better diffusion prior, but the deterioration when the score is not well approximated is more graceful than DPS (i.e., in less represented parts of the dataset). This can be useful when using a general inverse solver (e.g. ImageNet or foundation model) and solving for an image from an unknown or poorly approximated distribution.
>
> To further illustrate our point, we also demonstrate that the same trend occurs even when the diffusion model weights are initialized from a pretrained model, but from a different distribution than the source image. In Figure 10 in the rebuttal PDF, the right hand block shows an inverse problem setting for reconstructing ImageNet images, using an FFHQ model. As expected, our proposed method outperforms DPS on this out-of-domain inverse problem setting.

---

> > ### Comment · Reviewer_u6BH · 2024-08-10
> > **Response to authors**
> >
> > Dear authors, thank you for the detailed response and the additional experiments. This clarifies most of my questions. I would like to keep my positive score unchanged.

---

### Official Review · Reviewer_GqZF · 2024-07-06

**Soundness:** 3
**Presentation:** 3
**Contribution:** 3
**Rating:** 7
**Confidence:** 4

**Summary:**

This paper proposes a new approach to conditional generation tasks through score-based diffusion models, with a focus on inverse problems.   As an alternative to using the likelihood $p(y | x_t)$ to guide the time-reversed SDE towards the posterior distribution, the authors reformulate this as an optimal control problem.    Starting from the ODE-based time-reversed flow for the unconditioned prior, the authors derive a controller based on the iLQR algorithm to guide the particles towards high posterior probability regions.    The authors provide theory to demonstrate that the optimal guidance coincides precisely with the desired conditional scores.  They demonstrate the method on a number of benchmarks including image inpainting and other inverse problems.

**Strengths:**

The paper is well written and very clear.   The method appears novel and addresses a legitimate challenge in conditional diffusion models.  As the authors acknowledge: optimal control formulation of diffusions exist, but not (to my knowledge) in the context of guiding conditional diffusion models.     The theoretical results provide a sound justification of the validity of the approach.   The numerical results demonstrate that it is competitive in terms of accuracy compared to baseline, established methodology.

**Weaknesses:**

The main weakness is the sheer computational cost of the algorithm,  the need to compute very expensive hessians drastically limits the practical use of this method.    The authors suggest a number of low rank approximations to mitigate this, but it is unclear how much is lost by introducing them.      One point of question is the interplay between the number of diffusion steps $m$ and $T$.   As $m\rightarrow \infty$, for $T$ fixed and large we expect that the baseline conditional diffusion model will improve significantly in accuracy.  Generally, I feel that the configuration of the baseline has not been explored (or if it has, it has not been reported carefully).   Similarly, the authors claim that they have done equivalent budget analysis in the experiments -- I could not find the details of this: is it the case that the computational cost is the equivalent?  Have the author really explored the hyper-parameter space for these methods.

**Questions:**

Can the authors provide some insight on how to choose the key parameters $(m, n, k)$?  -- does the optimal control method allow substantially smaller $m$?   When does one approach become more computationally effective than the other?   I can imagine, when $m$ is sufficiently small, that PSLD, DPS will start to outperform this method with comparable computational cost.

Minor comments:  the metrics LPIP, PSNR, SSIM are reported, but at no point are these explained, or are references provided.  These are well known in some communities, but not for the wider readership.   Small typos around references were found.

**Limitations:**

The main limitation is the large computational cost of this methodology.   This has been identified and acknowledged.  The authors have claimed to do an equivalent budget analysis, but this maybe needed a bit more details (wall-clock time, etc).   Potential societal impacts are addressed in the Appendix.

---

> ### Author Rebuttal · Authors · 2024-08-07
>
> We thank the reviewer for their positive assessment of our work, and their insightful critique. Below we provide a point-by-point response to the reviewer's discussion points.
>
> **Main weakness is the computational cost of the algorithm, e.g., computing Hessians. Approximation error is not well understood.**
> Indeed, the Hessians would be very difficult to compute in their entirety. However, we want to mention that we do investigate the computational trade-off that occurs when approximating the Hessian $V_{xx}$. Our primary method technique is the randomized low rank SVD (Halko et. al, 2011), which provides very strong approximation guarantees when there does exist low rank structure in the matrix. Indeed, the Hessians of overparameterized neural functions have been well known to be low rank (Sagun et. al, 2017), and we also verify this empirically in Table 4 in the Appendix, where we see that the algorithm performance does not deteriorate with sparser representations, in terms of rank. Moreover, even letting the rank $k = 1$ obtains very strong performance (see also Table 1). We generally found in our experiments that a low-rank approximation is sufficient to impose the quadratic trust-region optimization to our model, and does not meaningfully deteriorate performance.
>
> **Understanding the interplay between the number of diffusion steps $𝑚$ and $𝑇$.**
> We generally find that, fixing a computational budget (e.g., 1000 NFEs), the number of diffusion steps and number of iLQR iterations can quite significantly affect the final outcome of the model. The key observation is that the iLQR objective optimizes strictly the inverse constraint, whereas the diffusion model acts as a regularizer. With a fixed budget, increasing one will come at the cost of the other. Therefore, if the measurement is very noisy, or the solution is known to be well-supported by the diffusion prior distribution, then one should take a large number of diffusion steps, and fewer number of iLQR iterations. Conversely, if the measurement is noise free, or the prior is not very informative, then the reverse should be done.
>
> **Equivalent budget analysis. Is the computational cost equivalent?**
> We perform equivalent budget analysis in two ways. First, we note that theoretically, our Algorithm 1 costs  $\mathcal{O}(nm(k^3 + kd^2))$ in terms of FLOPS and $\mathcal{O}(mnk)$ in terms of neural function evaluations (NFEs). Perhaps interestingly, we found that NFEs dominate the wall-clock time of the algorithm, accounting for 97%. We let $k=1$ in all reported experiments outside of our ablation study in Table 4. More details can be found in Appendix D.2.
>
> Empirically, the results of our equivalent budget analysis is summarized in the Ours (T=20) entry Table 1. As discussed, neural network calls take up ~97% of the wall-clock runtime of our solver. Therefore, we measure computational budget by the number of neural function evaluations (NFEs) of the method. DPS, MCG, DDNM, and DDRM allow a computational budget of 1000 NFEs, and most other methods in Table 1 follow a similar guideline. Therefore, the entry “Ours (T=20)” runs with 50 iLQR iterations and T=20, satisfies an equivalent budget to these models. We see that under this restricted budget our method beats out several prominent methods, including DPS, while comparing favorably against other state-of-the-art methods.
>
>
> **Have the author really explored the hyper-parameter space for these methods. Can the authors provide some insight on how to choose the key parameters (𝑚,𝑛,𝑘)?**
> Please see Tables 4, 5, and 6 for an in-depth exploration of the main hyperparameters of our method.
>
> We assume the reviewer is using the notation from Appendix D.2 for $(m, n, k)$. Fixing a computational budget $\mathcal{N}$, there is a clear trade-off between the number of diffusion steps $m$ and iLQR iterations $n$, since they are inversely related. More diffusion steps bias the model towards the diffusion prior, whereas more iLQR iterations bias the model towards the observation $y$.
>
> **When $m$ is small, DPS and PSLD should outperform this method with comparable cost.**
> We in fact observe the reverse phenomenon with $m$! In other words, our model actually outperforms DPS at low $m$. We conduct the simple comparison below on the super-resolution task on FFHQ. We fix $n = 50$, and report LPIPS.
> | Algorithm / $m$| 1 | 5 | 10 | 25 | 50 | 100 |
> |-|-|-|-|-|-|-|
> | Ours | 0.643 | 0.472 | 0.277 | 0.185 | 0.171 | 0.169 |
> | DPS | 0.927 | 0.799 | 0.485 | 0.395 |  0.354 |  0.331 |
>
> **The metrics (e.g., LPIPS, PSNR, SSIM) are mentioned but not described.**
> Thank you for pointing this out. We agree that this is can be confusing and have added references to the LPIPS, PSNR, and SSIM metrics.

---

> > ### Comment · Reviewer_GqZF · 2024-08-11
> > **Response**
> >
> > I thank the authors for their detailed response, and clarifications which were insightful.
> >
> > Taking into account the new information provided, I will increase my score.

---

### Official Review · Reviewer_KZ9Z · 2024-07-07

**Soundness:** 2
**Presentation:** 3
**Contribution:** 2
**Rating:** 5
**Confidence:** 3

**Summary:**

The paper uses the optimal control theory to solve the diffusion posterior sampling problem by iterative Linear Quadratic Regulator (iLQR) algorithm. The method could be utilized to solve both linear and nonlinear inverse problems.  Experiments on MNIST and FFHQ demonstrate the outperformance of the proposed method.

**Strengths:**

1. This paper is well-written, with a good summary of previous methods and their shortcomings.

2. The proposed method that interprets the reverse diffusion process as an uncontrolled non-linear dynamical system is novel. Theoretical support is provided to verify the algorithm.

**Weaknesses:**

1. The method is well-backed but might be computationally exhausting.

2. The experiments are limited. Quantitative results on different datasets and nonlinear inverse problems are lacking.

**Questions:**

1. As shown in Algorithm 1, the method's time complexity is  $O(T)$ in each iteration. Although the $T$ is relatively small $(=50)$ in the experiments,  num_iters $\times T$ would be large, e.g. $50\times 50 = 2500$ as shown in Table 2 in the appendix.  Also, the initialization of $\{x_T'\}$ requires $T$ NFEs (number of function evaluations). Is this correct?  How about the computational efficiency of the proposed method? I would like to see a more detailed comparison of the method with other baselines like DPS in terms of time.

2. More baselines need to be compared such as [1], [2], [3] and [4]. The settings in these works might be a bit different. Some settings might be different. Can you clarify the proposed method's advantage over these baselines?

3. Previous methods like DPS have done extensive experiments on both linear and nonlinear inverse problems across both FFHQ and ImageNet datasets. However. the experiments in the paper seem to be somewhat limited. I have two questions about the experiments. 1) Since there are only quantitative results for linear inverse problems (note that the results in Table 1 are all linear), can you clarify the proposed method's advantages in nonlinear problems such as phase retrieval, nonlinear deblurring, and so on? 2) Can you show more results on ImageNet, which is a broader dataset that contains more than one domain, such as faces in FFHQ?

[1] Zehao Dou, and Yang Song. Diffusion Posterior Sampling for Linear Inverse Problem Solving: A Filtering Perspective, ICLR 2024

[2] Morteza Mardani, Jiaming Song, Jan Kautz, Arash Vahdat. A Variational Perspective on Solving Inverse Problems with Diffusion Models. ICLR 2024

[3] Jiaming Song, Arash Vahdat, Morteza Mardani, Jan Kautz. Pseudoinverse-Guided Diffusion Models for Inverse Problems. ICLR 2023

[4] Zhu, Y., Zhang, K., Liang, J., Cao, J., Wen, B., Timofte, R., and Van Gool, L. Denoising diffusion models for plug-and-play image restoration. CVPR 2023

**Limitations:**

Yes. They mentioned limitations and impacts in the appendix, which looks good to me.

---

> ### Author Rebuttal · Authors · 2024-08-07
>
> We thank the reviewer for their appreciation of the optimal control perspective, and for their insightful discussion and bringing many recent works to our attention. We respond to comments in detail, on a point-by-point basis below.
>
> **The method is well-backed but might be computationally exhausting.**
> We maintain that our method is not significantly more expensive than competing methods. For details, see computational complexity analysis in Section D.3, where we show that iteration-for-iteration, our method has similar costs to DPS. Indeed, under the $T=50$ configuration, our method will run for more iterations. Moreover, under the $T=20$ configuration in Table 1, our method has a very similar runtime as DPS (130s vs 125s on an NVIDIA A6000 GPU).
>
> **Quantitative results on different datasets and nonlinear inverse problems are lacking.**
> Thank you for this suggestion. We have added experiments on ImageNet 256 x 256-1K and the phase retrieval and nonlinear blurring inverse problems, and find that our algorithm compares very favorably against existing methods.
>
> **Algorithm 1 takes 2500 NFEs in the $T=50$ case in Table 1.**
> Indeed, the reviewer is correct. We see that reporting only $T$ may not show the full picture, and have updated Table 1 to show NFEs. The new Table 7 (see main rebuttal PDF) displaying ImageNet results already shows $NFEs$.
>
> **What is the computational efficiency of the proposed method?**
> While the full matrix computations in equations 19-25 would be expensive to compute in full, we find that low rank approximations are very effective and result in minimal deterioration of the algorithm performance (for more details, see Appendix D.3 and specifically Table 4). Moreover, we find that the algorithm has the same computational complexity as most existing algorithms (e.g., DPS), and is similarly dominated by neural network calls (i.e., 97% of the wall-clock time is dominated by NFEs). Therefore, we also use NFEs to measure runtime.
>
> **I would like to see a more detailed comparison of the method with other baselines like DPS in terms of time.**
> Under the $Ours (T=20)$ configuration in Table 1, we take the number of iterations to be $50$, resulting in $1000$ NFEs. Under this equivalent computational budget to other methods in Table 1, we show that we still outperform prominent methods such as DPS and DDRM, while comparing favorably to state-of-the-art algorithms. Finally, we verify that the computational budget is indeed equivalent by comparing our method to DPS and MCG, letting $T = 1000$. On an NVIDIA A6000 GPU, all three methods take ~120s to run.
>
> **Please discuss advantages over [1], [2], [3] and [4].**
> We thank the reviewer for bringing these works to our attention. After careful study, we note that [1, 3, 4] all rely on Tweedie’s formula to predict an estimated $x_0$, given $x_t$, to formulate the correction step at each time of the reverse diffusion process. Therefore, these methods all suffer from the same pitfall of DPS (and MCG, and other algorithms of the same vein), where $\nabla log p(y | x_0)$ must be approximated.
>
> We do observe that [2] stands out from the other three in that does not require this approximation. However, the variational framework that is proposed is itself intractable in their inner optimization loop, and they require a stop-gradient on the score network to maintain tractability. Overall, these related works are important to discuss, and we have added these references to our manuscript.

---

> ### Comment · Reviewer_KZ9Z · 2024-08-13
>
> I thank the authors for their hard work on the rebuttal and appreciate the additional results. However, I still have concerns about two points:
>
> - Insufficient baselines: While I agree that DPS-like baselines share the same issues due to the use of Tweedie's formula, recent works like  FPS have shown significant performance improvements. Additionally, Resample (https://arxiv.org/abs/2307.08123, ICLR 2024) is an advanced method of PSLD, as both approaches are latent-based.
>
> - Unsatisfactory results: In Table 1 of the paper, the proposed method (T = 20) does not outperform either PSLD or DPS and is not even competitive. I wouldn't expect a significant difference between the FFHQ (Table 1 in the paper) and ImageNet datasets(in the rebuttal pdf) to change this outcome.
>
> Given these considerations and other reviewers' comments, I retain my score.

---

### Official Review · Reviewer_d6zE · 2024-07-09

**Soundness:** 3
**Presentation:** 3
**Contribution:** 3
**Rating:** 6
**Confidence:** 3

**Summary:**

This paper tackles inverse problem via the perspective of optimal control. By treating the diffusion process (ODE) as a non-linear dynamic system and the extra guidance term as control signal, the authors manage to optimize the diffusion trajectory via the iterative Linear Quadratic Regulator (iLQR) algorithm. Several techniques are used to make the iLQR algorithm more efficient. This paper show good results on FFHQ dataset.

**Strengths:**

The idea is interesting and reasonable. Using optimal control to solve the inverse problem enables us to optimize the whole sampling trajectory and avoid the error for estimating $x_0$ via Tweedie's formula. And the results on FFHQ dataset looks good.

**Weaknesses:**

1. High computation cost: Despite the advantages mentioned above, one obvious drawback of this method is the potential high computation cost. This includes:

      a. Computing and storing the Jacobian matrices, which can be of very high dimension, can be very costly. Although the authors
      further propose some techniques to reducing the cost, these methods might also bring extra approximation error as well as more hyper-parameters to tune;

     b. Optimizing the the whole trajectory requires evaluating the whole trajectory for many times and do iterative updates. This requires more computation. Thus, though in Table 1, the authors denoted their methods as $T=50$ and $T=20$, considering the iterative update nature over the whole trajectory, this might not be directly comparable (and might actually need more computation) to other methods, which are denoted as $T=1000$. And the authors might have to greatly reduce the timesteps to make the whole algorithm affordable, this might also bring extra approximation error.


2. Lack of more complex dataset: Though the authors achieve good performance on FFHQ dataset, considering the human face data is relatively easy (aligned, not very multimodal), it is still not very clear to me how the proposed method can work on more complex dataset, for example, on ImageNet. From my own experience, the ImageNet data can be much harder than the FFHQ human face data in the inverse problem. And considering the approximation error introduced in iLQR algorithm, computing the Jacobian matrices as well as using less timesteps, it might raise concerning regarding whether the proposed algorithm can work well on more complex dataset.

3. Minor suggestion: I think it might be better for the authors to add more introduction for the optimal control part in the main paper. Or at least give more clear introduction for the notation used in 2.3. Currently, I find it not very clear to people without much background in optimal control.

**Questions:**

1. Following my first point in weakness, can the author provide a comparison in sampling time (e.g. second, or NFE) of different methods. Only comparing diffusion timesteps are not very fair considering the proposed method needs to iteratively update over the whole trajectory for many times.

2. Under different initializations, can the proposed algorithm always be able to find a good solution? And will the optimized results look same or different?

**Limitations:**

The authors has adequately addressed the limitations.

---

> ### Author Rebuttal · Authors · 2024-08-07
>
> We would like to express our gratitude that the reviewer appreciates both the theoretical and empirical results of our work, and for bringing several insightful shortcomings of our work to our attention. Below, we address the reviewers concerns on a point-by-point basis.
>
> **High computational cost:**
>
> **a) Jacobian matrix computation can be costly, and low rank approximations bring additional error.**
> Indeed, Hessian matrices in iLQR are costly at full rank. Moreover, we agree that low rank approximations generally introduce some error to the computation. However, we maintain that this error is negligible. In our ablation study, we see that even a rank-one approximation of these matrices is sufficient, with further increases in rank providing quickly diminishing returns (see Table 4 in Appendix). Using modern randomized linear algebra libraries (e.g. randomized SVD), this brings the computation of the Hessian matrices to cost $\mathcal{O}(d)$, where $d$ is the data dimension. Therefore, the cost of our algorithm is on par with competing methods, e.g. DPS. (For more details, please find a thorough complexity analysis in Appendix D.2.) We also find that there are few hyperparameters introduced by these approximations, and the sensitivity of the algorithm to these hyperparameters is low (Tables 4, 5, and 6).
>
> **b) The proposed algorithm is iterative over the simulated trajectory. Reported $T$ in Table 1 may not be directly comparable.**
> The reviewer is correct, our Algorithm 1 requires multiple passes through the diffusion trajectory. Our intention with displaying the time steps $T$ in Table 1 is to show the stability of our algorithm even at very low time steps. This is a nice property to have, and can be useful in certain cases. For example, DPM-Solver (Lu et. al, 2022) is known to be unstable at >100 timesteps, and is usually used for T ~ 20. Methods like DPS will struggle in this regime.
> That being said, we do agree with the reviewer that in terms of computation time, the number of neural function evaluations (NFEs) is also important to consider, and have added it to Table 1, as well as the new Table 7 in the rebuttal PDF. Overall, we find that our model performs favorably against other methods under an equivalent computational budget (the T=20 case in Table 1 — and 1000 NFEs), and establishes a new baseline at T=50 and 2500 NFEs.
>
> **More complex dataset than FFHQ, e.g., ImageNet.**
> We agree that further experiments can demonstrate the generalizability of our results. To this end, we evaluate our model on the ImageNet-1K dataset replicating the experimental setup in (Chung et. al, 2022), and on some nonlinear forward operators. Please see the general rebuttal and PDF.
>
> **Minor suggestion: More introduction for the optimal control part in the main paper.**
> We fully support this suggestion. As the reviewer has noticed, we moved a large amount of the introduction to the appendix due to page constraints. However, following this suggestion we have included a slightly more intuitive discussion on iLQR and its mechanism in Section 2.3.
>
> **Only comparing diffusion timesteps is unfair. Please provide a second form of comparison for sampling time.**
> We agree, and again note that the “Ours (T=20)” entry in Table 1 is run for 50 trajectory iterations, yielding 1000 NFEs. This results in roughly the same sampling time (e.g., on an A6000, DPS and our method both take ~120s to run). We also maintain the same 1000 NFE budget in the new Table 7 showcasing results on ImageNet.
>
> **Robustness to different initializations.**
> We currently initialize our actions as identically 0, and our states as i.i.d. normal (as prescribed by the DDPM algorithm). In the rebuttal Figure X we show the result of the algorithm with different state and action initializations. We currently do not fix the seed of the state initializations, and our algorithm obtain similar results on each run. Therefore we find our method to be relatively robust to different initializations.

---

> > ### Comment · Reviewer_d6zE · 2024-08-11
> >
> > I would like to thank the authors for their responses. After reviewing the rebuttal pdf and comments, my concerns are resolved and I would like to raise my rating.

---

### Official Review · Reviewer_VcWe · 2024-07-19

**Soundness:** 1
**Presentation:** 2
**Contribution:** 2
**Rating:** 4
**Confidence:** 3

**Summary:**

The paper uses tools from optimal control to introduce a novel approach for solving inverse problems with diffusion models. The authors propose reframing the generative process of diffusion models as a discrete optimal control problem allowing to leverage the iterative Linear Quadratic Regulator (iLQR) algorithm. Tackling limitations of existing probabilistic sampling methods, the resulting method demonstrates promising performance for inverse problems on FFHQ, such as super-resolution, inpainting, and deblurring.

**Strengths:**

While many connections between optimal control and diffusion models have been established, the proposed algorithm leverages variants of iLQR to provide a fresh perspective on training-free posterior sampling with diffusion models. The paper provides additional theoretical guarantees as well as multiple modifications (randomized low-rank approximations, matrix-free evaluations, and adaptive optimizers) to reduce computational costs. Finally, several ablations are presented for the proposed method.

**Weaknesses:**

1. The claims of the paper are not sufficiently supported by experiments and/or theory (the first statements in the following are just examples---similar statements can be found throughout the paper):
	* "dependence on the approximation quality of the underlying terms in the diffusion process": only a result for a single image is provided (Fig. 6). The current paper also does not seem to provide theoretical results for such robustness as claimed in "reconstruction performance is theoretically and empirically robust to the accuracy of the approximated prior score".
	* "its sensitivity to the temporal discretization scheme": for the baselines, again only a result for a single image is provided (Fig. 3). Moreover, the number of steps is typically reduced to accelerate the algorithm. Accordingly, methods should be compared in terms of performance vs. runtime/flops and not the number of diffusion steps. It seems that the proposed method is significantly more expensive than competing methods (in particular, since `num_iters>=50` full simulations are used).
	* "its inherent inaccuracy due to the intractability of the conditional score function": The conditional score function remains intractable, one just obtains an approximation via iLQR, since the obtained $x_0$'s obtained from the iLQR iterations only *approximately* converge to the posterior distribution *in the limit*. Statements like "Moreover, our model always estimates $x_0$ exactly, rather than forming an approximation $\hat{x}_0 \approx x_0$" sound misleading. Using iLQRs, we simulate "nominal" trajectories and thus iteratively obtain an approximate candidate for $x_0$ which will be used for the refinement of the control. In a similar (however, useless) fashion one could also use, e.g., DPS to obtain an estimate of $x_0$ and then run a probability flow ODE simulation where the scores are conditioned on $x_0$ (instead of $x_t$) to have a "method [that] produces controls that coincide precisely with the desired conditional scores". However, the advantage of DPS lies in the fact that only a single simulation is needed.
	* "on several inverse problem tasks across several datasets": Apart from a single figure on MNIST (without metrics and for only a single baseline and task), results are only provided for FFHQ.

2. Moreover, several of the mentioned limitations have been already tackled by alternative approaches to posterior sampling with diffusion models, e.g., variational approaches (https://arxiv.org/abs/2305.04391) or resampling strategies (https://arxiv.org/abs/2307.08123).
3. Finally, the appendix could provide further details on
	* hyperparameter choices and optimization for the baselines.
	* precise assumptions for the theorems.

**Questions:**

See "weaknesses" above.

**Limitations:**

See "weaknesses" above.

---

> ### Author Rebuttal · Authors · 2024-08-07
>
> We thank the reviewer for their helpful discussion and insightful critique. We hope to clarify some of the points of the paper, and alleviate the reviewer's concerns below in a point by point response.
>
> **Only a single a single image is provided in Figs. 3 and 6.**
> In the rebuttal PDF Figures 9 and 10, we provide further experiments that demonstrate the concepts in Figure 3 (accuracy of the predicted score) and Figure 6 (robustness to the approximated score), respectively.
>
> **No theoretical results for such robustness [to the score approximation].**
> This theoretical statement comes from simple fact that DPS (and related algorithms) sample from p(x|y) by solving the conditional reverse diffusion process (Eq. 1), where $\nabla \log p_t(x_t)$ is approximated by the neural score function $s_\theta$ (equivalently $\epsilon_\theta$, $v_\theta$, etc. in noise- and v- prediction models, respectively, up to a simple reparameterization). Therefore, the performance of DPS (and related algorithms) depends on the approximation $s_\theta \approx \nabla \log p_t(x_t)$. Our algorithm makes no such assumption, and is thus robust to this approximation error. We further illustrate this point with more examples in the rebuttal Figure [x].
>
> **Methods should be compared in terms of performance vs runtime / flops and no the number of diffusion steps.**
> We do run an experiment under an equivalent runtime/flops budget in the Ours (T=20) case in Table 1, where we restrict our method to 1000 NFEs, we still show significant gains over DPS, and comparable performance against the most recent state-of-the-art methods on FFHQ256-1K. Empirically, we find that our method has a similar runtime to DPS on an A6000 GPU (130s vs 125s). That said, we do agree that listing only the number of diffusion steps can misrepresent our method. Therefore, we have added NFEs to Table 1 (just like Table 7 in the rebuttal PDF).
>
> **The proposed method is significantly more expensive than competing methods.**
> We maintain that our method is not significantly more expensive than competing methods. For details, see computational complexity analysis in Section D.3, where we show that iteration-for-iteration, our method has similar costs to DPS. Indeed, under the $T=50$ configuration, our method will run for more iterations.
>
> **The conditional score function remains intractable in the proposed method.**
> We respectfully disagree. Let us first follow DPS (Chung et. al, 2022) and define the conditional likelihood as $p(y | x_0)$. Therefore, the conditional score function at time t is $\nabla_{x_t} \log p(y | x_t)$. Unlike DPS, we consider both deterministic and stochastic dynamics in our theory.
> Under deterministic dynamics, we first note that the conditional **likelihood** is tractable, since $p(y | x_t) = p(y | x_0)$ — i.e., $x_0$ is determined by $x_t$, by definition — and $p(y | x_0)$ can be obtained via the forward rollout of Algorithm 1.
> Now, we have the distinct advantage over DPS where $\nabla_{x_t} \log p(y | x_0)$ **is** exactly backpropagated to $x_t$ (i.e., the backward pass of Algorithm 1, see Theorems 4.1-4.3). For this reason, we also maintain that the conditional **score** is tractable. This claim cannot be made by DPS, because 1) DPS is derived from stochastic dynamics and 2) DPS relies on Tweedie’s formula at each step to “approximate” the gradient of the conditional likelihood.
>
> **The statement that our model estimates $x_0$ exactly, rather than forming an approximation sounds misleading.**
> We also respectfully disagree here. We do estimate $x_0$ exactly. Note that $x_0$ as defined in DPS (as well as our work) is $x_0 | x_t$. In other words, the denoised image **given the current noisy image $x_t$** at time $t$. Observe that we *do* obtain this term exactly in each iLQR step, up to the discretization error of the diffusion solver.
> That said, we have added the qualification “up to the discretization error of the numerical solver” where applicable. We note that we do emphasize the discrete nature of our method in the abstract and throughout the paper, such as lines 67-68, 115, and 265.
>
> **With DPS one could also (uselessly) obtain an ODE estimate of $x_0$ and then compute the scores from there.**
> Indeed, this is possible --- but like the reviewer says, uselessly slow. The impracticality of this strategy illustrates the advantage of our method. Using iLQR, we are able to combine the prediction of $x_0$ in the forward rollout with the score computations used in the feedback step in one single sweep (Algorithm 1). To see this, observe that the scores can be computed for all timesteps with a single forward solve (and backprop) of the ODE. This is not possible for DPS, since each conditional score *must be* computed sequentially, **AFTER** the previous score has already been applied to the trajectory. If DPS were to use the reviewer’s suggested strategy, it would require on average $n / 2$ extra NFEs per update, resulting in a sum total of 500K NFEs under the current parameterization. By comparison, our method achieves superior performance to DPS using as little as 1K NFEs.
>
> **Results are only provided for MNIST and FFHQ.**
> We additionally provide experiments on ImageNet-1K, and demonstrate that our performance holds, with similar trends to Table 1.
>
> **Please discuss alternative approaches to posterior sampling with diffusion models, e.g., variational approaches (https://arxiv.org/abs/2305.04391) or resampling strategies (https://arxiv.org/abs/2307.08123).**
> We thank the authors for bringing up these concurrent works. We have added them to the manuscript. We note that ReSample (Song et. al, 2024) suffers from the same approximation error as DPS when estimating $x_0$ via Tweedie’s. Conversely, RED-Diff (Mardani et al., 2024) indeed does not require Tweedie’s, but requires approximations to their variational framework (e.g., the stop-gradient) when computing their proposed variational loss, thus also yielding an approximate solution.

---

> ### Author Response · Authors · 2024-08-07
> **Rebuttal (Continued)**
>
> **Hyperparameter choices for the baselines.**
> For baselines from competing works in Table 1, we directly replicate the reported hyperparameters from the respective papers. For our baselines, please refer to Table 2 for the hyperparameter values. For the T = 20 result, we let T = 20 and let the number of iterations be 50. For insight into how we selected these values, please see Section D.3 and Tables 4, 5, and 6.
>
> **Assumptions for the theorems.**
> The assumptions required for Theorem 4.1 are summarized in Equations 27 and 28. Similarly, Theorem 4.2 requires Equations 30 and 31. We have additionally clarified that we require $\alpha > 0$, and $\log p(y | x)$ to be twice-differentiable (which holds when $p(y | x)$ is Gaussian, as is the case in our settings). We take our theoretical results very seriously and would be happy to clarify any other ambiguities in the theorems.

---

> > ### Comment · Reviewer_VcWe · 2024-08-12
> >
> > **No theoretical results for such robustness [to the score approximation]:** "Therefore, the performance of DPS (and related algorithms) depends on the approximation $s_\theta \approx \nabla \log p_t(x_t)$. Our algorithm makes no such assumption, and is thus robust to this approximation error." It is clear (and also written in 196-197), that the theoretical results require such an approximation. Thus, it still remains unclear *why* the method is more robust.
> >
> > **Methods should be compared in terms of performance vs runtime / flops and no the number of diffusion steps. / The proposed method is significantly more expensive than competing methods. / Results are only provided for MNIST and FFHQ.** I thank the authors for the additional experiments which improve the empirical evaluation and resolve some of my concerns. A revised version of the paper should provide results for different NFEs to assess the scaling as well as provide more insights into runtime for all considered method (considering the maximal batchsize for each method for a given memory budget). However, when comparing against further baselines, such as FPS-SMC [1], DCDP [2], DiffPIR [3], RedDIFF [4], ReSample [5], the provided results for ImageNet, in particular for the deblurring tasks, seem not to be state-of-the-art anymore.
> >
> > **The conditional score function remains intractable in the proposed method.** The conditional likelihood (also the one used for the ODE) is defined via $p(y|x_t) = \int p(y|x_0) p_{SDE}(x_0 | x_t) \mathrm{d}x_t$, where the conditional density of the SDE $p_{SDE}$ cannot just be replaced by the one for the ODE (which, I agree, would be a dirac, since for deterministic dynamics $x_0$ is exactly determined by $x_t$.).
> >
> > **The statement that our model estimates $x_0$ exactly, rather than forming an approximation sounds misleading.**
> > Indeed one can just simulate the SDE, however, the distribution of $x_0$ depends on the approximation of the score (see also the first comment).
> >
> > **Please discuss alternative approaches to posterior sampling with diffusion models, e.g., variational approaches (https://arxiv.org/abs/2305.04391) or resampling strategies (https://arxiv.org/abs/2307.08123).** Note that resampling-based methods, such as DiffPIR and DCDP, use a simulation instead of Tweedie's formula to obtain an estimate of $x_0$.
> >
> > **Assumptions for the theorems.** It should be defined what exactly is meant by $x_t$ (does it originate from the approximate score, the ground-truth score, the forward/backward simulation), on which equation (29) depends. Also, equation (29) comes without any statement.
> >
> >
> > Given the additional empirical evidence, I adjusted my score. However, I still think that the presentation should be improved, both theoretically (unclear statements, see above) as well as empirically (compare to further baselines, see above).
> >
> > ---
> >
> > [1] Dou, Zehao and Song, Yang. "Diffusion Posterior Sampling for Linear Inverse Problem Solving: A Filtering Perspective.", ICLR 2024.
> >
> > [2] Li, Xiang, et al. "Decoupled data consistency with diffusion purification for image restoration.", arXiv preprint arXiv:2403.06054.
> >
> > [3] Zhu, Yuanzhi, et al. "Denoising Diffusion Models for Plug-and-Play Image Restoration.", CVPR workshop NTIRE 2023.
> >
> > [4] Morteza Mardani, Jiaming Song, Jan Kautz, and Arash Vahdat. A variational perspective on solving inverse problems with diffusion models. arXiv preprint arXiv:2305.04391, 2023.
> >
> > [5] Bowen Song, Soo Min Kwon, Zecheng Zhang, Xinyu Hu, Qing Qu, and Liyue Shen. Solving inverse problems with latent diffusion models via hard data consistency. ICLR, 2023

---

> > > ### Author Response · Authors · 2024-08-13
> > >
> > > We greatly appreciate the reviewer's continued discussion of our paper, and their more positive outlook. Below we respond to some additional points the reviewer raised.
> > >
> > > **No theoretical results for such robustness [to the score approximation] -- see Lines 196-197.** Theoretically, the reasoning for such robustness is due to the simple fact that DPS (and other algorithms) differentiate through Tweedie's formula, which relies explicitly on the score. On the other hand, our algorithm differentiates through the true simulated forward dynamics, which can be any dynamical system --- this additional flexibility is granted by our generalized optimal control framework.
> > >
> > > **Comparison to ICLR 2024 results.** Indeed, the ImageNet results from ICLR 2024 (and some other venues) are very relevant, and we shall include some of these concurrent results in our work, where applicable. We note that some works do not use the same experimental setting. For example, ReSample uses the much easier setting of $\sigma = 0.01$, whereas we keep the settings used by DPS $\sigma=0.05$. We still believe that our work provides a meaningful contribution to the field, and hope the reviewer agrees.
> > >
> > > **The conditional score remains intractable.** We claim tractability of $\nabla_{x_t} \log p(y | x_t)$ in the deterministic case. We also claim tractability of $\nabla_{x_t} \log p(y | x_0)$ in the stochastic case, which DPS (and related models are still unable to compute). For details, see our response to *No theoretical results for such robustness [to the score approximation]*. We have made this clearer in the manuscript.
> > >
> > > **Estimating $x_0$ requires approximation of the score.** There is a subtle distinction here. $x_0$ as defined by the reverse diffusion process requires the score, of course. $x_0$ as defined simply as the final state of the dynamical system solved by our system does NOT require the score. We will make this distinction more clear in our manuscript.
> > >
> > > **Please discuss alternative approaches to posterior sampling with diffusion models, e.g., variational approaches (https://arxiv.org/abs/2305.04391) or resampling strategies (https://arxiv.org/abs/2307.08123).** We thank the reviewer for bringing these papers to our attention. DiffPIR actually uses Tweedie's (See Line 3 in Algorithm 1). On the other hand, DCDP does not leverage Tweedie's. However, there is also no theoretical discussion of the correctness of the proposed approach.
> > >
> > > **Assumptions for the theorems.** $x_t$ is defined as in Algorithm 1. We are missing a statement before Eq. 29, which should read: "Then the iterative linear quadratic regulator with Tikhonov regularizer $\alpha$ produces the control". We apologize for the confusion, and have added statements clarifying this in our manuscript.

---

### Author Rebuttal · Authors · 2024-08-07

We thank all reviewers for their thoroughness and diligence in reading our manuscript. We received a lot of sound, constructive criticism and positive feedback. This guided our revisions and further experiments in this rebuttal period, and we believe that the paper has meaningfully improved as a result. Below, we summarize some common assessments (good and bad) shared by multiple reviewers. We then address each reviewer's concerns on a point-by-point basis below.

**Strengths:**

All reviewers noted the novelty of our application of the well-known iLQR algorithm in optimal control theory to diffusion modeling. Reviewers VcWe, KZ9Z GqZF, and L2Ru found the theoretical contributions sound and compelling. KZ9Z, GqZF, and u6BH found the writing generally clear, and all reviewers provided suggestions that further improved the readability of our work.

**Weaknesses:**

**Lack of further empirical experiments.**
Many reviewers felt that, while performance is very strong on FFHQ, it is important to validate the performance of our algorithm on other datasets and tasks so as to understand the generalizability of the approach. We considered this feedback and extend the empirical results of our algorithm on two fronts: **ImageNet experiments** and **further nonlinear tasks**. In both settings, we constrain the number of function evaluations (NFEs) of our algorithm to 1000, in line with other works in these settings.

In Table 7 and Figure 7 in the attached rebuttal PDF, we demonstrate very favorable performance on ImageNet compared to existing works. We note that some works (e.g. PSLD) cannot be applied to ImageNet, since there do not exist unconditional latent ImageNet models, and are therefore absent from our benchmark.

In Table 8, we compare our model to DPS on two nonlinear inverse problems, phase retrieval and nonlinear deblurring following the setup in DPS (Chung et. al, 2022), and find that we improve on the baselines set by DPS on the phase retrieval task, and are on par with DPS on the nonlinear deblurring task.

Finally, we further corroborate our claim that our proposed algorithm is more robust to approximation errors in the neural score function. We demonstrate an out-of-domain inverse problem setting, where the model is tasked to recover an image that is out-of-distribution to the training distribution of the pretrained model. In Figure 10 in the rebuttal PDF, the right hand block shows an inverse problem setting for reconstructing ImageNet images, where the diffusion model is initialized with a pretrained model on the FFHQ dataset. As expected, our proposed method outperforms DPS on this out-of-domain inverse problem setting.

Overall, these results round out the empirical evaluation of our work, demonstrating that our proposed algorithm is capable of obtaining high quality solutions to inverse problems on a bevy of datasets and problem settings.

**Concerns on computational cost.**
Reviewers were also concerned by the computational cost of the algorithm. Indeed, optimal control was designed for generally simpler, low-dimensional systems, and many computations in a vanilla iLQR algorithm are  the Hessians would be very difficult to compute in their entirety. However, we want to mention that we do investigate the computational trade-off that occurs when approximating the Hessian $V_{xx}$. Our primary method technique is the randomized low rank SVD (Halko et. al, 2011), which provides very strong approximation guarantees when there does exist low rank structure in the matrix. Indeed, the Hessians of overparameterized neural functions have been well known to be low rank (Sagun et. al, 2017), and we also verify this empirically in Table 4 in the Appendix, where we see that the algorithm performance does not deteriorate with sparser representations, in terms of rank. Moreover, even letting the rank $k = 1$ obtains very strong performance (see also Table 1). We generally found in our experiments that a low-rank approximation is sufficient to impose the quadratic trust-region optimization to our model, and does not meaningfully deteriorate performance.

Many also noted that we only report $T$, not $NFEs$ in Table 1, which can be misleading about the runtime of our algorithm. Our original intention with displaying the time steps $T$ in Table 1 is to show the stability of our algorithm even at very low time steps. This is a nice property to have, and can be useful in certain scenarios. For example, consistency (Song et. al, 2023), progressively distilled (Salimans et. al, 2022) or adversarially finetuned diffusion models (e.g., Imagine-Flash, Kohler et. al), or even regular diffusion models sampled with certain solvers (e.g. DPM-Solver, Lu et. al, 2022) are known to be unstable at larger $T$. Methods like DPS, PSLD, MCG, etc., will thus struggle in this regime, and fail to find good solutions to the inverse problem at hand. That being said, we do agree that listing only $T$ is a somewhat unfair representation of our algorithm, and have adjusted the notation in Table 1 in line with Table 7 in our rebuttal PDF. In particular, Ours ($T=20$) now reads Ours (NFE = 1000) and Ours ($T = 50$) now reads Ours ($T = 2500$), and all other methods in Table 1 also list NFEs, where applicable.

---

### Comment · Area_Chair_Z5yw · 2024-08-09
**Please review the authors' rebuttal and respond**

Dear Reviewers,

Thank you once again for your time and effort in reviewing for NeurIPS 2024.

The authors have already submitted their rebuttal to your comments, and the discussion phase has now begun. For those who have not yet responded, please review the authors' responses carefully at your earliest convenience. Your timely feedback on whether your previous concerns have been adequately addressed is crucial to the reviewing process.

Thank you very much!

Best regards,
Area Chair

---

### Decision · Program_Chairs · 2024-09-25

**Decision:**

Accept (poster)

**Comment:**

After reading the reviews comments and the original paper,

Main strengths:
1. Novel idea. This manuscript proposed an optimal control perspective on solving linear inverse problems with diffusion models, which is an interesting and novel idea. The use of iLQR is very interesting and well-motivated.
2. Good empirical results. The empirical results show that the proposed method can achieve impressive reconstruction performances in terms of FID, LPIPS, as shown in Table 1.

Main weaknesses:
1. Theoretical arguments not sound and rigorous.
2. Insufficient comparisons. More baseline methods and data should be incorporate to validate the method. Moreover, while FID and LPIPS are good in Table 1, the performances in terms of SSIM are much worse compared to other methods.

Overall, given that the idea of using optimal control is novel and the performance is relatively good, I recommend acceptance but the authors should address the concerns raised in the rebuttal and discussions.